

# Ocean 2D eddy energy fluxes from small mesoscale processes with SWOT

Elisa Carli[1], Rosemary Morrow[1], Oscar Vergara[2], Robin Chevrier[2], and Lionel Renault[1]

[1]Laboratoire d'Etudes en Géophysique et Océanographie Spatiales (CNES-CNRS-IRD-UPS), Toulouse, France
[2]Collecte Localisation Satellites (CLS), Toulouse, France

**Correspondence:** Elisa Carli (elisa.carli@legos.obs-mip.fr)

**Abstract.** We investigate ocean dynamics at different scales in the Agulhas current system, a region of important interocean exchange of heat and energy. While ocean observations and some of the most advanced climate models capture the larger mesoscale dynamics (>100 km), the smaller-scale fronts and eddies are underrepresented. The recently launched NASA-CNES Surface Water and Ocean Topography (SWOT) wide-swath altimeter mission observes the smaller ocean geostrophic scales down to 15 km in wavelength globally. Here we will analyze different eddy diagnostics in the Agulhas Current region and quantify the contributions from the larger mesoscales observable today, and the smaller scales to be observed with SWOT. Surface geostrophic diagnostics of eddy kinetic energy, strain, and energy cascades are estimated from modelled sea surface height fields of the MITGCM / LLC4320 simulation subsampled at 1/10°. In this region, the smaller scales (<150 km) have a strong signature on the horizontal geostrophic strain rate, and for all eddy diagnostics in the western boundary current and along the meandering Agulhas Extension. We investigate the horizontal cascade of energy using a coarse-graining technique, and we observe that the wavelength range where the inverse cascade occurs is biased towards larger mesoscale wavelengths with today's altimetric sampling. We also calculate the projected sampling of the eddy diagnostics under the SWOT swaths built with the CNES-NASA simulator to include the satellite position and realistic noise. For the swaths, a neural network noise mitigation method is implemented to reduce the residual SWOT random error before calculating eddy diagnostics. In terms of sea surface height (SSH), observable wavelengths of 15 to 20 km are retrieved after neural network noise mitigation, as opposed to wavelengths larger than 40 km before the noise reduction.

## 1 Introduction

The Southern Ocean (SO) is a highly energetic region with an important role in our global climate, as it drives a large meridional overturning circulation and is a key region of the formation and modification of mode, intermediate and deep waters (Bourgeois et al., 2022). The cold and oxygen-depleted surface water masses allow efficient storage of the anthropogenic heat and carbon, making the SO the greatest sink for these tracers (45 - 62% of global ocean heat gain from 2005 to 2017) (IPCC, 2019). This SO circulation includes large-scale mean and highly turbulent transient flows, due to eddies, fronts and meanders (Morrow and Le Traon, 2012; Sallée, 2018), which is the most energetic component of the global ocean circulation (Wunsch, 2002; Chelton et al., 2007). The energetic eddy component of the SO circulation is key in transporting heat poleward across the





zonally circulating Circumpolar Current (Volkov et al., 2010). Horizontal and vertical fluxes induced by eddies and meanders contribute to the redistribution of energy, momentum, and tracers such as heat, carbon and nutrients (Thomas et al., 2008; Lévy et al., 2012; Sallée, 2018). SO eddy kinetic energy varies regionally in relation to the mean currents and orography, but also evolves interannually in response to climate modes and wind-forcing, with eddies rectifying the mean flow. (Meredith and Hogg, 2006; Morrow et al., 2010; Sinha and Abernathey, 2016; Martínez-Moreno et al., 2019). Areas of strong eddy surface
strain are also hotspots of smaller-scale (<50 km)ageostrophic ocean vertical movements in the SO (Su et al., 2018; Siegelman et al., 2020), with geostrophic surface strain impacting the vertical distribution of heat, carbon and nutrients (Zhang et al., 2019).

Within the SO, the Agulhas Current system, flowing southwestwards along Africa's continental shelf break, plays a key role in this global ocean circulation, feeding the upper branch of the Atlantic meridional overturning circulation (AMOC). From
the southern tip of the continental shelf (20°E), the current presents an abrupt curvature back into the South Indian Ocean known as the Agulhas Retroflection (Lutjeharms, 2006). Large Anticyclonic eddies known as the Agulhas rings are shed from the retroflection into the eastern boundary of the South Atlantic and contribute to the inter-basin exchange by leaking saline warm waters from the Indian Ocean to the Atlantic, (Olson and Evans, 1986; Gordon et al., 1992; Lutjeharms, 2006; Beal et al., 2011). The Agulhas Current System is one of the most energetic regions of the global oceans and presents a series of
large-scale (> 150 km) eddies and meanders, thought to be the main driving mechanism of the current until a few decades ago (Boyd, 1994). This mesoscale variability plays a strong role in the region's intense air-sea interactions (Speich et al., 2007; Swart et al., 2008; Renault et al., 2017) and in the inter-ocean exchange and water mass modification (Lutjeharms and Gordon, 1987; Gordon et al., 1992; Boyd, 1994; Speich et al., 2007).

Most of these studies are based on eddy-resolving simulations (1/12° or 1/10° spatial resolution) or global altimetry maps,
that resolve the larger mesoscales >100 km in diameter, but not the smaller mesoscales or submesoscales. Yet the ocean fine-scale dynamics (15 to 150 km), with their strong gradients in ocean properties and temporal scales ranging from days to weeks, are now understood to affect the ocean physics and biomass up to climate scales (Ferrari and Wunsch, 2008). Over the last decade, high-resolution ocean models have made great advances in resolving finer spatial and temporal scale dynamics, down to a few km regionally (Renault et al., 2018; Verron et al., 2020; Contreras et al., 2023; Bendinger et al., 2023), and globally
(Wang and Menemenlis, 2021; Arbic et al., 2022). These models provide a more detailed description of the dynamics of energetic western boundary current systems, in particular, the role of meso- and submesoscale structures and their rectification on the mean circulation (McWilliams, 2008; Gula et al., 2014; Renault et al., 2016; Chassignet and Xu, 2017; Renault et al., 2017; Contreras et al., 2023). Within the Agulhas Current System, a large number of high-resolution modelling studies have addressed the role of eddy-current interactions on the generation of the large "Natal Pulses" (Krug and Penven, 2011), of
coastal Kelvin waves (Sebille and Leeuwen, 2007), or of the sub-mesoscale fast barotropic motions (Tedesco et al., 2019; Schubert et al., 2020).

Although observational efforts such as the Argo program and remote sensing programs providing global observations of the large-scale ocean circulation and larger mesoscale ocean dynamics, a gap exists concerning global observations of the finer scales (< 150 km in wavelength) that could lead to misinterpretation or loss of key physical or biophysical mechanisms





in ocean models (d'Ovidio et al., 2019). Local in-situ campaigns can target individual, small-scale features in 3D, but they only represent a fraction of the ocean conditions, with limited time and spatial coverage. Satellite sea surface temperature (SST) and ocean colour observe these finer scales in non-cloudy conditions, which limits their applicability over the SO. Alongtrack satellite altimetry can resolve locally spatial scales down to 30 to 70 km when averaged through wavenumber spectral analyses (Arbic et al., 2013; Dufau et al., 2016; Renault et al., 2019; Vergara et al., 2023). Yet a significant part of the

observed sea surface height (SSH) variability originates at scales shorter than the 150 km in wavelength observed by altimetric maps today (Sasaki et al., 2014; Callies et al., 2015). Due to this observational gap, it is difficult to accurately observe the ocean's energy cascades. We are missing observations of how and where the small scales interact with the large-scale field, and to understand where the energetic small scales compensate or enhance the larger-scale flow. Indeed, larger mesoscale eddies often appear and disappear in mid-ocean (Chelton et al., 2011) since we are not correctly observing their key small-

scale generation and dissipation processes. Altimetric maps merging multi-multi mission information also underestimate the eddy energy in energetic boundary current regions, due to the spatial resolution/coverage trade-off of the mapping technique that smoothes out features smaller than 150 km. These are all open questions but need more fine-scale observations to close the global ocean energy budget (Ferrari and Wunsch, 2008). The recently launched NASA/CNES Surface Water and Ocean Topography (SWOT) mission is expected to partially fill this gap, by increasing the current observational capabilities of satellite

altimetry down to 15 km wavelength (Desai, 2018; Morrow et al., 2019). The improved SAR-interferometric technology will allow us to observe the fine-scale geostrophic sea surface height at 2 km resolution, a signature that responds to dynamics driven from within and below the surface mixed layer. SWOT is expected to provide an unprecedented data set to validate the geostrophic part of eddy-resolving ocean models down to scales of 15 km in wavelength (ie equivalent eddy diameters of 8 km, Rossby radii of 4 km).

As for all satellite observations, the observing capabilities of SWOT will be impacted by different sources of measurement error, due to instrument and atmospheric effects, and also the geophysical characteristics of the ocean circulation (waves, tides, etc) (Dibarboure et al., 2022). SWOT was successfully launched in December 2022 and the first data should be available in October 2023. In this paper, our objective is to diagnose SWOT observations in terms of signal and noise, in order to better understand the mesoscale dynamics observed today with multi-mission maps from DUACS/CMEMS in the Agulhas Current

system, and investigate the new dynamical scales to be revealed by SWOT. To reproduce SWOT we use a realistic global high-resolution model including tidal forcing, based on the Massachusetts Institute of Technology general circulation model (MITgcm) / LLC4320 simulation (Wang and Menemenlis, 2021). Modelled SSH fields and their geostrophic approximation for surface currents and surface eddy diagnostics are analysed in the Agulhas Current region. We also perform equivalent nadir altimetric sampling of our model SSH fields and then reconstruct 2D "pseudo-DUACS" maps to quantify the scales observable

with today's mapping and interpolation schemes. Various eddy diagnostics are calculated on both data sets, to compare the ocean dynamics resolved in today's observations, with those to be resolved by SWOT. These include the smaller scales < 150 km in wavelength and the important cross-terms within the energy cascade linking the smaller and larger-scale dynamics. We then diagnose the impact of the instrumental and geophysical errors on the SSH observations by applying the SWOT ocean simulator (Gaultier et al., 2016). An innovative artificial intelligence (AI) processing (Tréboutte et al., 2023) has been



developed and tested in the North Atlantic Ocean to reduce the impact of random instrumental noise in SWOT observations
and is planned to be implemented in the global SWOT operational environment. We will test its performance in reducing the
SWOT random noise in the Agulhas Current system, and assess the adaptability of this AI method on a different model and
different zone with respect to its training. Finally, we can estimate the scales of observability of the Agulhas Current system's
ocean dynamics in the ideal error-free scenario, for noisy SWOT data and after noise reduction.

The paper is organized as follows. The data and methods are introduced in Section 2. Section 3 presents the results of
the eddy diagnostics over the full Agulhas Current system model domain. In section 4 we present the results of the eddy
diagnostics extracted under the SWOT swaths, and compare the observability of the ocean eddy diagnostics before and after
the AI treatment of the residual KarIn random error. Finally, a discussion and a conclusion are provided in Section 5.

## 2   Methodology and data

This section describes the Massachusetts Institute of Technology general circulation model (MITgcm), which is the mesoscale
resolving model used for the analysis, the reconstructed pseudo-DUACS data used as a reference for today's altimetry mea-
surements capability, and the SWOT simulator used to reconstruct SWOT orbits and simulate the observed position and errors.

### 2.1   LLC4320 simulation

The model data used in this study to represent the total ocean signal is the output of the mesoscale and submesoscale resolving
MITgcm model (Marshall et al., 1997). The SWOT community has largely used the specific simulation LLC4320, where
'LLC' refers to a Latitude-Longitude-polar-Cap global model grid (Forget et al., 2015), which in its higher resolution has a
nominal horizontal grid spacing of $1/48°$ globally, similar to the SWOT swath grid spacing of 2 km x 2 km, with an effective
resolution of less than 20 km (Rocha et al., 2016). (Rocha et al., 2016) Outputs are hourly snapshots that span the period from
13 September 2011 to 15 November 2012 (Lin et al., 2020). There are 90 vertical levels ranging in thickness from 1 m at the
surface to 480 m at a maximum model depth of 7000 m. The LLC4320 is forced at the surface with 6-hourly 0.14°atmospheric
fields from the European Center for Medium Range Weather Forecasts (ECMWF) and by the 16 most significant components
of the hourly tidal potential, applied as additional surface pressure, generating both barotropic and baroclinic tidal components
(Chaudhuri et al., 2013; Wang and Menemenlis, 2021). A few details on the validation of the model for the specific use of this
study are given in Annex A. Further information about the LLC4320, its construction and validation can be found in Torres
et al. (2018).

This work focuses on sea surface height and geostrophic currents that would be observed with satellite altimetry in the
Agulhas region, represented in Figure 1, and our diagnostics are based on the modelled SSH. To reproduce an equivalent
altimetry-like SSH field, the model's barotropic tides and the high-frequency barotropic response to the atmospheric forcing
(the so-called dynamical atmospheric correction: DAC) are estimated and removed. The $1/48°$ simulation is known to have
minor forcing errors on the tides and an offset on the atmosphere forcing fields (by 6 hours) Arbic et al. (2022). This results
in barotropic and baroclinic tidal artefacts and wind-forcing offsets on the SSH signal. Unrealistic grid-type features at the



spatial scale of the atmospheric forcing are also observed in the 1/48° SSH field, creating discontinuities when deriving eddy diagnostics from the geostrophic components of the velocity field.

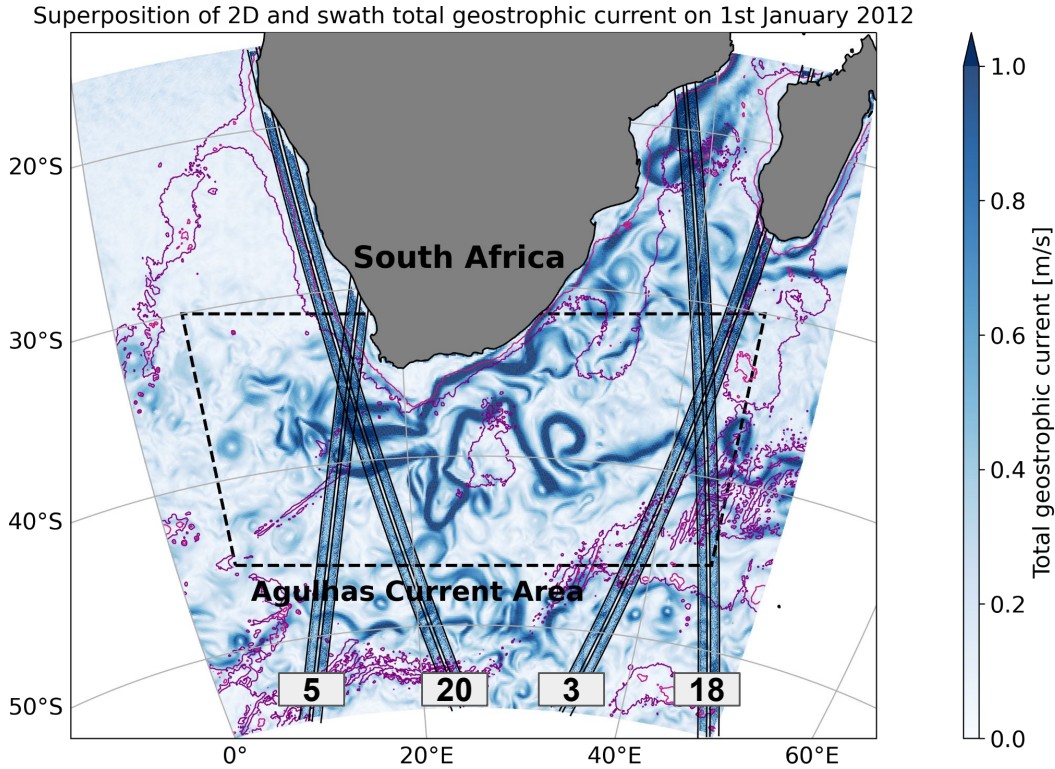

**Figure 1.** Superposition of 2D geographics map and swath total geostrophic current on January 1st 2012 in the Agulhas current region. The black dashed line defines the zone of this study. The yellow and purple contours define the 1500m and 3000m topography respectively. Data on the swath include instrumental and geophysical measurement errors. In the fast-sampling phase of the mission, four tracks will sample the region: 20, 5, 3, 18.

The data used in this work is the output of the original high-resolution run of the MITgcm LLC4320, smoothed on a regular
global grid at 1/20°, and sub-sampled at 1/10°. This allows us to greatly reduce the discontinuities previously mentioned. Since we do not aim to analyse the ocean geostrophic dynamics at scales smaller than the SWOT resolution (around 15 km at best), we will rely on the 1/10° sub-sampled modelled simulations for our study. This 1/10° version was also used to calculate the pseudo-DUACS dataset and for simulating the global SWOT cross-calibration. This dataset will be referred to as LLC10 in the following.



## 2.2 DUACS and pseudo-DUACS reconstructed data from LLC10 simulation

Today's multi-mission Data Unification and Altimeter Combination System (DUACS) altimetry products deliver global and regional sea level and geostrophic current maps for oceanographic applications (distributed by CMEMS). The latest version, the level-4 DUACS-DT is constructed by optimal interpolation of level-3 altimeter observations (i.e. with alongtrack corrections, editing, and processing applied) and is produced on a regular grid of $1/4°$ x $1/4°$ grid with daily sampling (Taburet et al., 2019). However, Chelton et al. (2011) found that the capability of retrieving small mesoscale dynamics in level-4 data is limited by the optimal interpolation framework used in DUACS. Ballarotta et al. (2019) analyzed the effective spatial and temporal resolution of level-4 SLA globally. For the time span of the LLC4320 simulation, the satellites in use for the generation of the DUACS product are Jason-2, ENVISAT, HY-2A and Cryosat-2 (Taburet et al., 2019). Further details on this observational product and its benchmarking against its surrogate derived from LLC10 can be found in Appendix A.

If we assume that our modelled SSH represents reality, our first step is to verify how this SSH would be observed by today's along-track altimetric sampling and mapping, so we built a pseudo-DUACS product. This product consists of the same DUACS mapping technique described above, however, the "data" are sub-sampled from the LLC10 fields along the altimeter mission tracks. These "altimeter-like" model data are then processed using the DUACS optimal interpolation mapping algorithm, using the same mapping parameters used for the real DUACS maps, providing pseudo-DUACS daily maps on a regular 1/10° grid. Since the original model has no data assimilation, its eddy field may be shifted compared to the real eddy position, and the advantage of the pseudo-DUACS is to have better colocation of the modelled large and small-scale features. So in this study, the Pseudo-DUACS product is used as a proxy for the current observational capability of satellite altimetry in our region, in terms of spatial and temporal resolution. In the region of the Agulhas current, the effective resolution of the pseudo-DUACS gridded maps is larger than 150 km and represents dynamics with temporal variability greater than 10 days. SWOT will potentially observe all scales included in the LLC10 SSH fields. In the computation, the smaller scale processes observable with SWOT have been defined as the residual between the total LLC10 fields (corrected for the high-frequency motions) and the Pseudo-DUACS fields. This derived product, referred to as "SSH residuals", will be analyzed to investigate the new dynamics that SWOT will observe.

Figure 2 illustrates the SSH spectral power of the different products derived from the LLC10 in terms of time and spatial frequencies, calculated over the annual model period. The top panels show the LLC10 original SSH (a) and the same field in (b) after removing the high-frequency correction for the barotropic tide and dynamical atmospheric correction (DAC). As expected, most of the higher frequency tidal energy (6 hours, 12 hours, 1 day) disappears in the corrected version of the spectrum. There are weaker residual energy signals at tidal frequencies that may be due to errors in the barotropic tide correction or at smaller scales, internal tides. Although the MITGCM LLC4320 is known to overestimate the tide forcing by a factor 1.1121 (Arbic et al., 2022), in our Agulhas Region, the residual internal tide signal is very low compared to the energetic mesoscale SSH field shown in Figure 2b. The DAC correction removes much of the large-scale, high-frequency energy at time scales < 10 days; since this has large-scale structure, it doesn't impact the eddy diagnostics we will derive that are shaped by the smaller-scale SSH gradients. The SSH spectral power for the reconstructed pseudo-DUACS product is shown in Figure 2(c), and for the SSH




residual high-frequencies in Figure 2(d). The time frequencies of the pseudo-DUACS product and the SSH residuals are cut

at the Nyquist frequency of two days because of their daily time sampling. We note that even if the pseudo-DUACS mapping

decorrelation scales are around 150 km in the Agulhas region (Ballarotta et al., 2019), its frequency-wavenumber spectra have

some residual power to scales smaller than the effective resolution of 150 km in 2c, and smaller lobes of energy around 30 km

in wavelength. Whereas the SSH residuals (d) represent well the residual SSH power (b-c) at scales < 150 km and between

2-10 days.

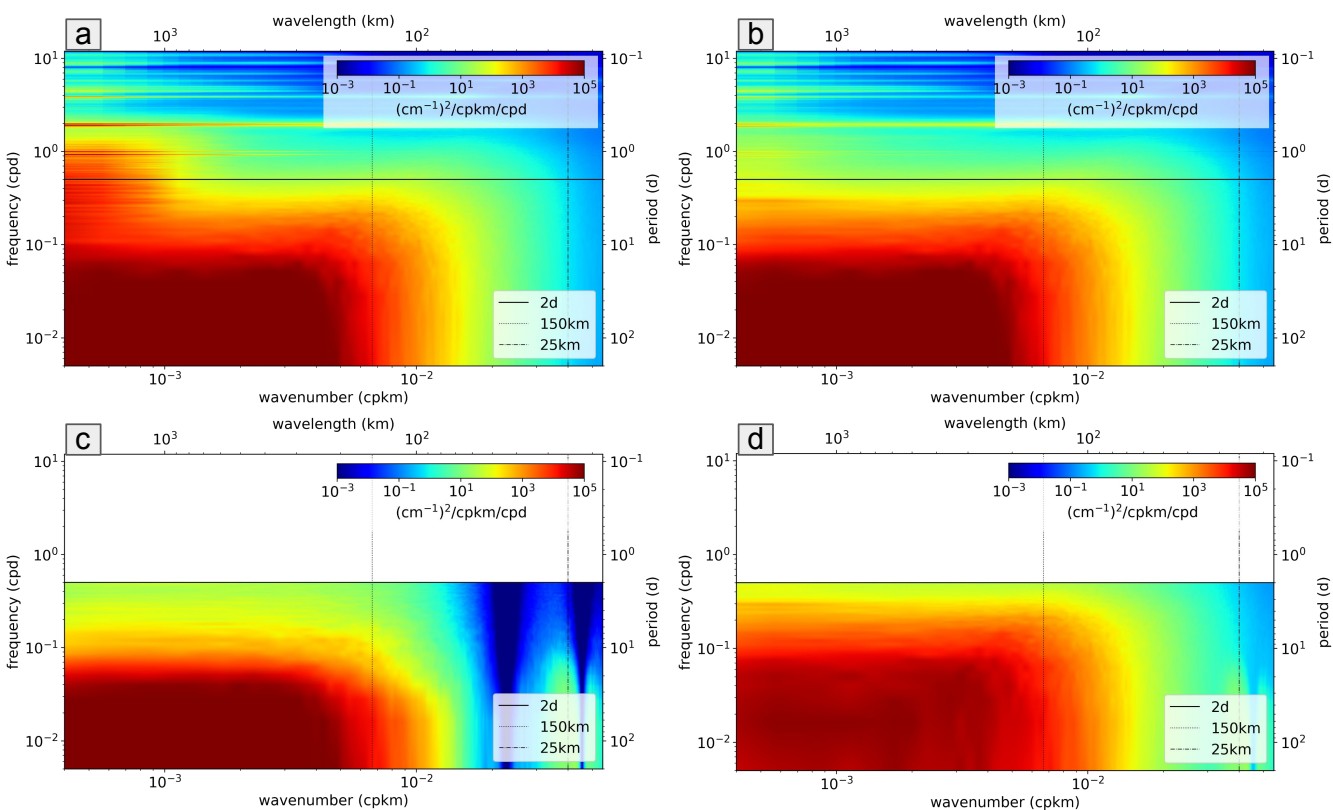

**Figure 2.** SSH wavenumber-frequency energy spectra. a) Total SSH from LLC10 simulation, b) SSH from LLC10 corrected for tide and DAC, c) large-scale pseudo-DUACS data on the LLC10 simulation, d) residual from b)-c) retaining the high-frequency and smaller scales SSH

**2.3   SWOT mission and the SWOT ocean simulator**

The Surface Water Ocean Topography (SWOT) mission, launched on 16 December 2022, will provide the first global SAR-interferometry observations providing colocated 2D surface topography and 250 m resolution SAR images (Fu et al., 2012; Morrow et al., 2019). SWOT observations are made over two 50 km wide swaths, with a Jason-class nadir altimeter in the centre. An onboard processor allows for a 250 m x 250 m expert product and a precise 2 km x 2 km grid product over all





oceans. The measurement is very precise, with an expected root mean squared (rms) SSH noise at 2 km of 1.37 cm, an order of magnitude smaller than conventional Jason-class altimeters (Chelton, 2019). SWOT will have two orbit phases: a daily revisit over a limited number of tracks during the first six months (January to June 2023) of the mission's commissioning and a Calibration and Validation phase (the Cal/Val orbit). From July 2023, the satellite moves into the science orbit with a 21-day revisit and global coverage. Both orbits cover up to latitudes of 78° (Dibarboure et al., 2022). SWOT is the first altimeter

mission whose science requirements are specified in terms of wavenumber spectra and observable wavelengths; these requirements are that the SSH signal should exceed the noise (for 2m wave height conditions) at 15 km wavelength globally (Desai, 2018). During the daily CalVal phase of the SWOT mission, a series of international in-situ validation campaigns are proposed (swot-adac.org). One of these campaigns "Quicche" is proposed in the Agulhas "Cape Cauldron" region (https://www.swot-adac.org/campaigns/quicche/) where a SWOT cross-over exists (SWOT tracks 5 and 20; Figure 1). This region is used as a test

site in our study to evaluate the projected SWOT signal-to-noise ratio.

### 2.3.1 SWOT simulator

The SWOT simulator provides the swath position and instrument errors along SWOT tracks, for both the daily fast sampling and the 21-day repeat science orbits as described by Gaultier et al. (2016). Here the main characteristics are given, for more in-depth information on the simulator please refer to https://swot-simulator.readthedocs.io/en/latest/. The SWOT simulator

allows the extraction of SSH data from an ocean general circulation model, and the interpolation of the latter onto the SWOT swath grid. For our study, the "signal" is derived from the LLC10. The simulator interpolates the original model grid onto the alongtrack-crosstrack SWOT 2 km grid, the SSH interpolation is linear in space, and no interpolation is performed in time. The results presented in this study are from the Cal/Val phase, so the outputs are daily cycles on the fast sampling orbit.

The SWOT simulator then allows us to generate most sources of error for the SWOT measurement, as foreseen in the

SWOT error budget documentation (Esteban-Fernandez, 2013) and described in Gaultier et al. (2016); Rodriguez et al. (2017). The main sources will be instrumental: KarIn random noise (which varies with the sea state and other inhomogeneities) and systematic errors (satellite roll, phase errors, baseline dilation, timing errors, orbital errors). There are also contributions from geophysical errors at mesoscales (mainly wet troposphere). The SWOT simulator uses a global spectral wavenumber estimate of each error and projects that back locally onto the swaths in space and time. Figure 1 is an example of the 4 daily fast

sampling orbits that pass in the Agulhas current region, showing a snapshot of the total geostrophic current computed with the SWOT simulator data (with the full spectrum of errors added to the data) over the LLC10 simulation. It is clear that the full spectrum of SSH error and their gradients makes it impossible to interpret the underlying geostrophic current field, and that the finer noise structures could be misinterpreted as mesoscale structures, as already observed by Fu and Ubelmann (2014), and Chelton et al. (2022). Dibarboure et al. (2022) estimate that the systematic errors alone contribute tens of centimetres in SSH.

This study focuses on the influence of the KarIn random error on the SSH data and observable wavelengths in SSH, eddy kinetic energy (EKE), and strain rate. The KarIn random error is defined as a Gaussian noise with a zero-centred distribution, and is strongly dependent on sea-state. As shown in Esteban-Fernandez (2013), its standard deviation depends on the distance to the nadir and on the significant wave height (SWH). Since a realistic SWH is not available in the LLC4320 simulation, the





KarIn random error for our analysis has been generated from the spectral SWOT requirements specification with a constant
SWH of 2 m.

### 2.3.2  Mitigation of KarIn random error

The step of de-noising along-track or swath altimetry data is fundamental for exploiting the real altimeter signal. In this work,
we only focus on the reduction of the random error associated with the KarIn instrument. The SWOT project will perform
cross-calibration to reduce the larger-scale errors (Dibarboure et al., 2022). Wang et al. (2019) computes the wavelengths
observable with SWOT when the residual random error is added to the SSH signal, based on a realistic wave field and no
noise reduction strategy. At higher latitudes with higher wave fields such as in the Southern Ocean, they find that without de-
noising, wavelengths smaller than around 40 km will be hidden by the random noise. The observability wavelength increases
as higher-order derivatives are computed on the SSH, such as geostrophic velocities (first-order derivative) or diagnostics like
the strain rate (second-order derivative). To address this problem of the random noise in a 2D field such as SWOT, several
methods have been explored, such as the spatial-based median and Lanczos filters (Fan et al., 2019), or the Boxcar, Gaussian
or Laplacian filters. Gómez-Navarro et al. (2020) and Gómez-Navarro et al. (2018) show that these spatial filters do not give
satisfying results. In Gómez-Navarro et al. (2018), a variational filter was specifically created to treat SWOT data. The objective
was to minimize a cost function by optimizing a parameter called $\lambda_2$ which depends on the study area, and the season, and
maintains smooth second-order derivatives (eg vorticity). Tréboutte et al. (2023) developed an alternative method based on
a convolutional neural network (CNN), called U-Net, being more efficient than previous filtering techniques, and preserving
small-scale ocean features, including near coasts and islands. In this work, we will evaluate the performance of the U-Net noise
mitigation technique to reduce the random noise on our pseudo-SWOT observations.

The de-noising model is based on a U-net architecture, which is trained and tested on the five years of the eNATL60
simulation (https://zenodo.org/record/4032732) in the North Atlantic region. The basic training of the U-Net is performed
by comparing the de-noised SSH to the true reference SSH using a loss function (Tréboutte et al., 2023). If the residual is
not improved for a chosen number of steps N, the training stops to avoid overfitting. For information on the noise reduction
technique, the reader is referred to Tréboutte et al. (2023). The robustness of the technique has been tested for different
scenarios, including a de-noising of the global dataset of the GLORYS model at 1/12°, based on training only in the North
Atlantic with the eNATL60 model.

Our objective is to test the U-Net de-noising algorithm in the region of the Agulhas current with the LLC10 model outputs
along SWOT tracks generated using the SWOT simulator. This dataset represents different dynamics from the North Atlantic
training zone, being in a different and very energetic zone, with a different model. The U-Net algorithm has not been retrained,
and here we test its capabilities of adaptability to a different, energetic region and a different model, in order to estimate if this
method will be applicable to the early real SWOT data.

The scores put in place to assess the robustness of the noise reduction technique are the following: the root mean square
error (RMSE), the variance of the SSH residuals, and the wavelength observable over the swaths. We compute the observability
following Wang et al. (2019), as the ratio between the spectral content of the noisy signal ($h_{noisy}$) and the spectral content





of the reference SSH ($h_{true}$), simulated via the SWOT simulator without added noise. In this work, the noisy signal is either the LLC10 SSH with the total added KarIn random error, or the LLC10 SSH with the residual noise after treatment with the U-Net. The observable wavelength is the wavelength for which the noise to signal ratio (NSR) reaches one. The two results will be compared to understand the influence of the U-Net denoising on SWOT observability.

$$NSR = \frac{PSD(h_{noisy} - h_{true})}{PSD(h_{true})} = 1 \tag{1}$$

where the $PSD$ is the power spectral density of the signal.

## 3  Diagnostics on the 2D maps

This section aims at showing the potential of the SWOT mission by analyzing eddy diagnostics based on the model simulated SSH alone, with no SWOT errors applied. We will compare the results for the larger mesoscales (>150 km) observed today, represented by the pseudo-DUACS product, and the smaller and faster dynamics that should be newly observed by SWOT, represented by the residuals of the LLC10 dataset after removing the pseudo-DUACS. Although SWOT aims to observe the full spectra of larger and smaller geostrophic mesoscale structures and their interactions, we will focus on the residuals and the smaller scales.

The main geostrophic eddy diagnostics investigated in this work are described below.

– Relative vorticity: diagnoses the SSH field variability in terms of its small-scale and turbulence content. We compute the normalized form as:

$$\xi = (\frac{\partial v_g}{\partial x} - \frac{\partial u_g}{\partial y})/f \tag{2}$$

where $u_g$ and $v_g$ are the surface geostrophic velocities derived from the altimetric SSH ($h$)

$$u_g = -\frac{g}{f}\frac{\partial h}{\partial y}$$
$$v_g = \frac{g}{f}\frac{\partial h}{\partial x} \tag{3}$$

where $g$ is gravitational acceleration and $f = 2\Omega sin\theta$ is the Coriolis frequency, proportional to $\Omega$, the Earth's rotation rate, and to $\theta$ the latitude.

– EKE: identifies eddy variability, and altimetric EKE is often used to validate the realism of ocean models. This component of the energy is directly related to the temporal evolution of the fluid parcels. It is computed from the anomalies of the zonal and meridional components of the geostrophic velocities. The anomalies are defined as the velocity minus its temporal mean at each grid point.

$$EKE = \frac{1}{2}(u_g'^2 + v_g'^2) \tag{4}$$





– Strain rate: an estimate of the surface deformation of the current. At the surface, regions of high strain are associated
with a secondary circulation characterized by high vertical velocities. These areas can be key for the vertical exchange
of heat, carbon, and nutrients and are biologically more productive (Zhang et al., 2019). Its formulation is defined from
surface geostrophic velocities.

$$S_g = \sqrt{\left(\frac{\partial u_g}{\partial x} - \frac{\partial v_g}{\partial y}\right)^2 + \left(\frac{\partial v_g}{\partial x} + \frac{\partial u_g}{\partial y}\right)^2} \tag{5}$$

– Energy cascades between different scales: Traditionally the fluxes of energy between different scales are computed with
a spectral method, or more recently with the coarse-graining method (Leonard, 1975; Germano, 1992), used for the first
time in oceanography by Aluie et al. (2017). The coarse-graining method is more efficient than the spectral method and
allows the generation of 2D maps of energy cascades at a fixed scale L (Schubert et al., 2020; Contreras et al., 2023). It
does not require windowing nor the hypothesis of an isotropic field, which avoids a large amount of data loss (and energy

loss). Here, the coarse-graining method has been used to study the geostrophic eddy fluxes in the Agulhas current with
the LLC10 simulation. Consequently, the wavelength limit considered is 10 km at 45°S, and smaller scales would not be
interpretable from the model and are below the scales observable with SWOT.

The term $\Pi$ representing the scale transfer of kinetic energy can be derived by convoluting the equation of motion:

$$\Pi = -\rho_0[(\overline{u^2} - \overline{u}^2)\overline{u}_x + (\overline{uv} - \overline{u}\,\overline{v})(\overline{u}_y + \overline{v}_x) + (\overline{v^2} - \overline{v}^2)\overline{v}_y] \quad . \tag{6}$$

A difficulty in this approach is related to the choice to be made close to continental boundaries. Here, we treated land as
water with zero velocity, as in Aluie et al. (2017).

### 3.1 Geographical distribution of EKE and strain rate

Our first objective is to analyze the different eddy diagnostics based on the larger and smaller mesoscale dynamics observable
with today's altimetric mapping and with the future SWOT observations, using the LLC10 as our ocean reality in the Agulhas
Current region. Figure 3 shows the geographical distribution of the average EKE and the strain standard deviation (std) over
the one-year period of the LLC10 simulation for three separate cases: the top row is calculated from the total model (LLC10)
geostrophic currents, computed from the SSH after correction for barotropic tide and DAC, and represents the total field to be
observed by SWOT; the middle row represents the results computed with the pseudo-DUACS data, a proxy of the capability
of products derived from current-generation altimeters to measure the larger mesoscale structures; the bottom data represents
their difference: the new scales to be observed with SWOT. We use it to tease out the circulation variability currently not
included in the operational products. In terms of mean surface geostrophic EKE over this 1-year period, the maximum for the
total corrected LLC10 current occurs in the Agulhas Retroflection (0.256 m²/s²) which is close to the reported observations
of the geostrophic EKE computed from DUACS-DT2021 data on CMEMS over a longer period (from January 1st 1993 to
June 3rd 2020, not shown). The larger scales in the pseudo-DUACS (Figure 3c) are responsible for most of the energy in the
Agulhas retroflection (0.171 m²/s²) compared to the smaller-scale residuals (0.085 m²/s²) (Figure 3e). The larger-scale energy





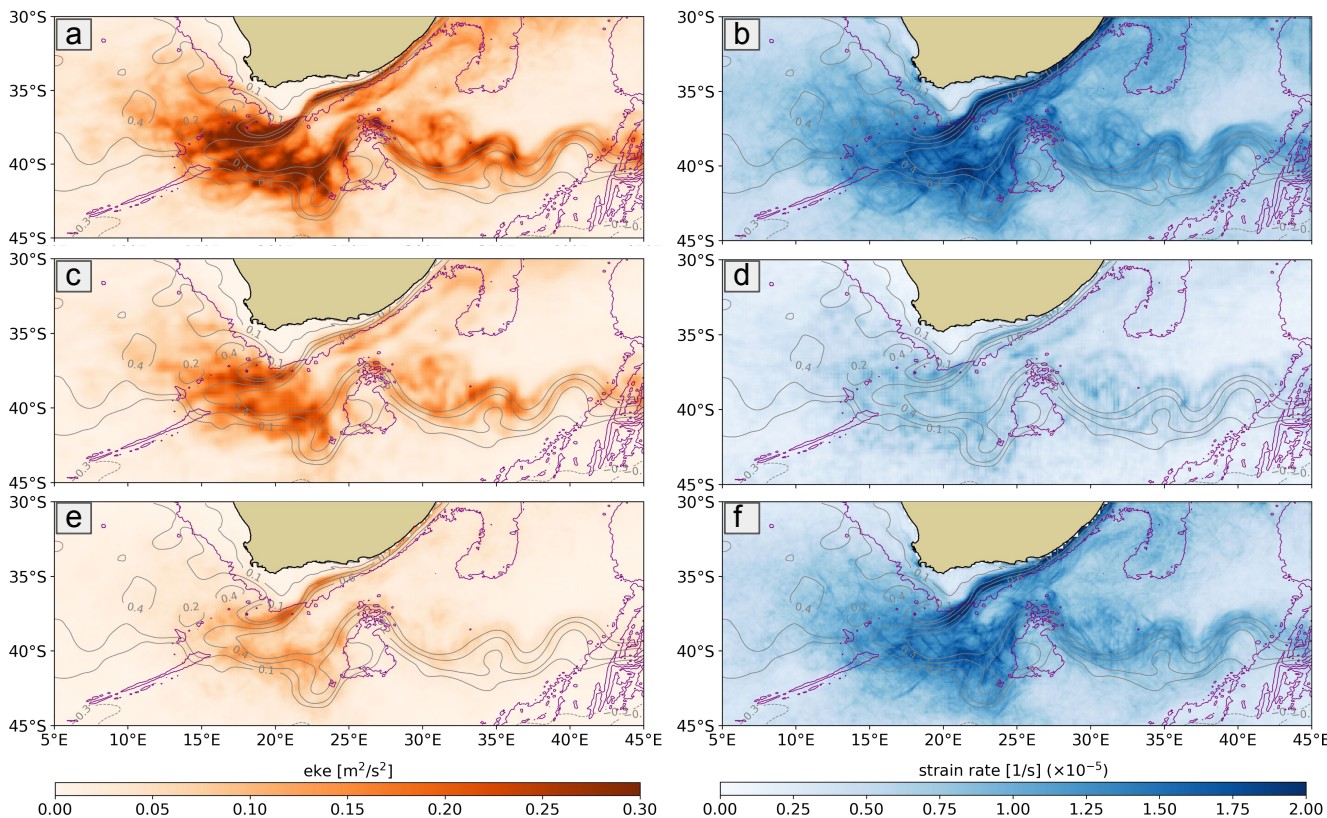

**Figure 3.** Average EKE (left) and strain rate standard deviation (right) over the full LLC10 simulation period (September 2011 to November 2012) (a, b), the pseudo-DUACS product (c, d), and the residuals between the simulation and the pseudo-DUACS product (e, f). The grey contours are the yearly mean current and the purple contours represent the 3000m bathymetry. Note that the residual small scales are computed as the difference between the LLC10 full model and the pseudo-DUACS dominated by the larger-scales (>150 km)

follows the meanders of the main current in the Agulhas extension. The LLC10 simulation represents physics down to 20 km (Figure 2b), whereas the pseudo-DUACS is limited by the interpolation scheme. The residual and smaller scales, representing the missing dynamics, have a weaker signature in EKE except along the Agulhas Shelf, especially near Port Elisabeth (around 26° E) where the LLC10 current dynamics present greater smaller-scale turbulence. In the CMEMS observations in Figure

A1a, the larger mesoscale develops further south, whereas the model develops weaker large-scale energy offshore, all along the Agulhas Current, with smaller-scales dominating on the shelf. This is characteristic of the DUACS interpolation that forces the reconstructed dynamics close to zero near the coasts and thus underestimates the boundary currents and their variability. The strain root-mean-square variations show a different behaviour. The mean annual strain structure is dominated by the larger scales (not shown). However, Figure 3 shows that the small-scale, high-frequency dynamics (f) are predominant in terms

of variance over the full Agulhas current system and that as an average over the year, the larger scales (d) are quite stable.





The majority of the small-scale strain variability comes from the coastal Agulhas Current and the Retroflection and loses its intensity along the Agulhas extension.

## 3.2   EKE and strain rate variability in four specific regions

As illustrated in Figure 2, the SSH variance in the Agulhas region is essentially dominated by the rich variability occurring at

synoptic to intra-seasonal timescales. In order to analyze the spatial coherency of such variability, we compute the Hovmöller diagrams of the LLC10 geostrophic EKE (Figure 4), focusing on the region delimited by the time-average current position (grey contours in Figure 3) along the Agulhas coast and extending to the Retroflection (orange line in the geographic plot of Figure 4). Four zones can be distinguished in the area: zone 1 is the northern part of the coastal region, from where the Agulhas current starts flowing southward in the study area. Zone 2 is the coastal extension south of Port Elisabeth. Zone 3 is

the retroflection of the current, and zone 4 is the Cape Cauldron region to the west where the QUICCHE CalVal campaign is planned. The first three zones are located along the mean current and are marked on the Hovmöller diagram, from which two sets of distinctive physical processes are distinguishable.

The first is associated with two strong, large mesoscale events called Natal Pulses, represented by the blue dashed lines in the plots. Natal Pulses are solitary meanders that develop near the Natal Bight (around 31° E) and can reach 50 to 200 km

in amplitude and have been largely studied with satellite altimetry (Schouten et al., 2002; Lutjeharms, 2006). They originate south of the Mozambique Channel and travel downstream, westwards towards the tip of the retroflection. Up until they reach the Agulhas Bank (around 24° E), continued processes of dissipation and merging occur (Krug and Penven, 2011; Krug and Tournadre, 2012; Krug et al., 2014) and only a fraction of the Natal Pulses reaches this point. The Agulhas current becomes increasingly unstable downstream and in addition to the Natal Pulses, shear-edge eddies develop (Lutjeharms and Gordon,

1987), with diameters between 50 and 100 km, that are difficult to observe with altimetry maps. During 2012 in the LLC10 model simulation, two Natal Pulses are reproduced in the coastal region of the Agulhas current, propagating at an average speed of 6 km/day. At the northern limit of the domain, they have a size of around 150 km, consistent with the observations and literature. Around mid-March, when the first Natal Pulse reaches Port Elisabeth, its structure is stretched along the mean current direction and can reach 300 km. The DUACS-reconstructed data, representing the potential of current altimetry, is

able to resolve these slow, large-scale processes (in a similar Hovmöller diagram (not shown)) in terms of diameter and phase, but their amplitude is underestimated by about 40%. This is a well-documented characteristic of the gridded products, as a consequence of the mapping algorithms (Ballarotta et al., 2019). Figure 4 highlights that the first Natal Pulse passed box 1 in Dec 2011, propagated at 6.3 km/day, reached box 2 in early March 2012, and then the retroflection box 3 in early May 2012. An animation shows that after passing the unstable zone of Port Elisabeth, part of the Natal Pulse is split into one Agulhas

Ring and propagates in the region of Cape Cauldron. The rest merges in the Agulhas Current and continues down through the Agulhas Extension.

The Hovmöller diagram highlights a second type of much smaller and faster dynamics that have time-varying amplitudes along the Agulhas Shelf from zones (1) to (2), and are almost always present in the zone of the Agulhas Bank, except in the periods when the large Natal Pulses pass. Similar dynamics are modelled in the Gulf Stream (Gula et al., 2016). We find that



these smaller-scale features propagate at a speed of around 20 km/day and have a diameter of a few tens of km, too small to
be observed with gridded altimetry maps. Tedesco et al. (2019) and Krug et al. (2017) demonstrate that southwards from Port
Elizabeth, barotropic instabilities are the main cause of smaller mesoscale eddies generation, in a period when no Natal Pulses
are occurring. Our LLC10 fields are in accordance, and in the proximity of Port Elisabeth the Natal Pulse loses most of its
energy and the flow becomes highly unstable generating faster smaller-scale eddies.

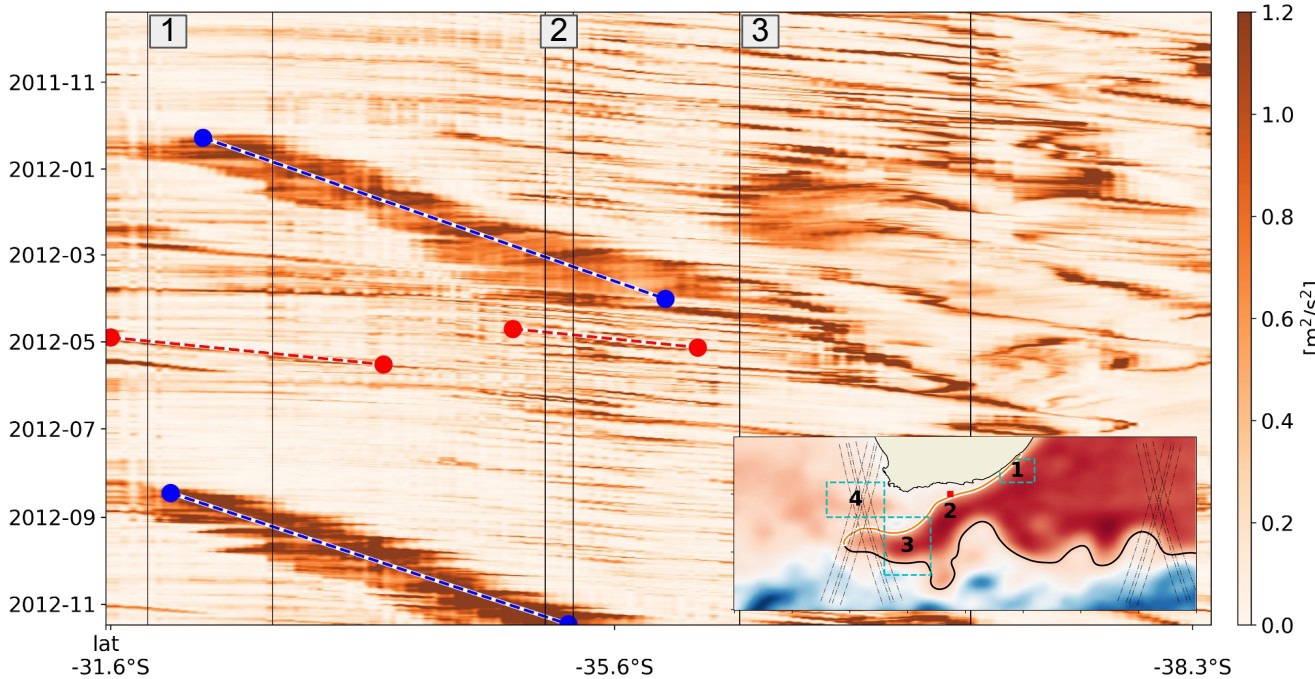

**Figure 4.** Hovmöller diagram of the Agulhas current coastal region EKE, computed from the *corrected SSH* of the LLC10 simulation. Zones
1, 2, and 3 define specific boxes, shown in the geographical section of the plot. The Y-axis represents the time and the X-axis represents the
distance from the first northern point, following the mean current shown in orange in the geographical plot of Figure 4. The blue lines show
the slower and larger scale dynamics, so-called Natal Pulses generated in the Mozambique channel, whereas the red lines represent the faster
and smaller-scale ones generated in the Agulhas Bank after the pass of the Natal Pulse.

We can quantify how well the different datasets represent these dominant large and small-scale processes by focusing on
the temporal evolution of the EKE for each of the four boxes defined in Figure 4. Figure 5 shows the EKE on the full period
of the LLC4320 simulation from the LLC10 data, from the pseudo-DUACS data and their residual. The mean EKE and strain
value for each box are in Table 1. In the highly energetic Agulhas region, the larger mesoscale represented by the pseudo-
DUACS product (orange line in Figure 5) is structuring the flow in the coastal box (a), but the large and small scales have
similar mean values. In the retroflection (c) and cape Cauldron (d) boxes the large scales dominate the EKE, with a mean value
close to double the small scales in both cases. Yet, in comparison with the total LLC10 data (blue line), the pseudo-DUACS is
underestimating the magnitude of the larger eddies in strong events, by 30-40%. The biggest features here are the two Natal



Pulses propagating downstream from the coastal box 1, along the shelf break (box 2) to the Retroflection area (box 3), as shown in Figure 4.

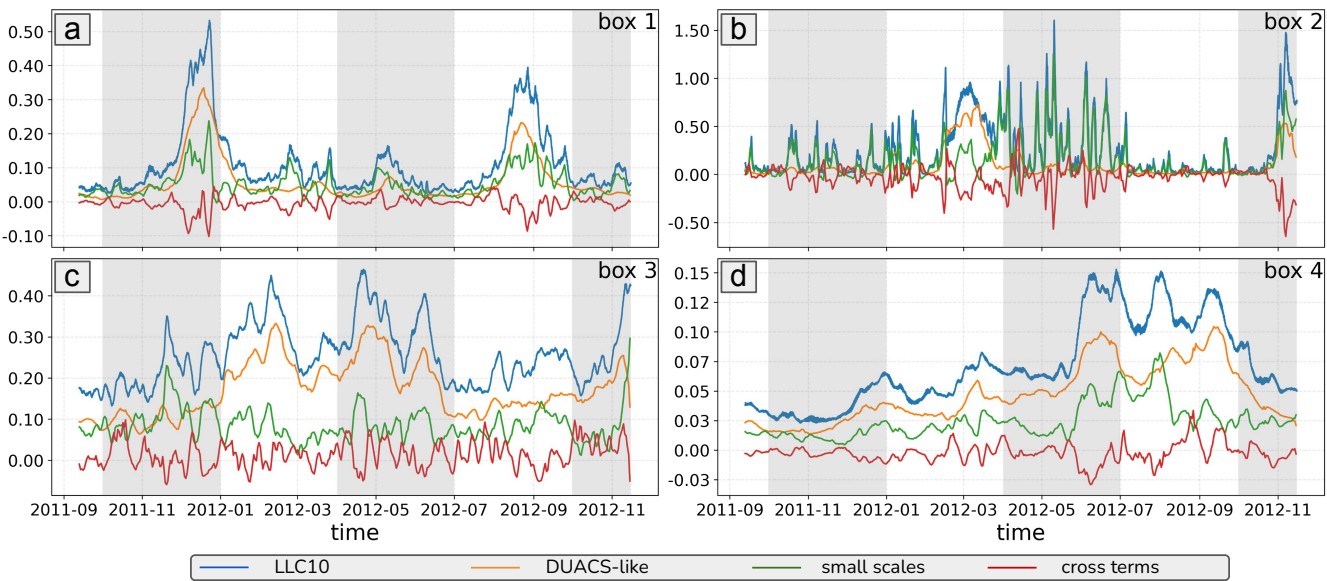

**Figure 5.** EKE [m$^2$/s$^2$] in the four regions defined in Figure 4: upstream coastal zone (a), coastal extension (b), Agulhas retroflection (c), and Cape Cauldron zone (d). The four superposed lines correspond to the full LLC10 simulation (blue), the pseudo-DUACS data (orange), the residual high-frequency smaller-scale (green), and the large/small scales cross terms (red)

In these three boxes, the amplitude of the smaller-scale dynamics generally increases in periods when the larger-scale eddies are active, suggesting that energy transfer processes are activated during this period. Certain smaller-scale energetic processes are smoothed-out by the pseudo-DUACS reconstruction, such as in November 2011 in box 3 and the large peak in late July 2012 in the Cape Cauldron region. Box 2 is different, being a shelf region dominated by small-scale, rapidly propagating dynamics that are stronger on average than the larger scales in the same region, except during the periods when the two Natal

**Table 1.** EKE and strain mean on the four zones identified in Figure 4

|  | LLC10 | | Pseudo DUACS | | Residuals | |
|---|---|---|---|---|---|---|
|  | eke | strain | eke | strain | eke | strain |
|  | [m$^2$/s$^2$] | x10$^{-5}$[1/s] | [m$^2$/s$^2$] | x10$^{-5}$[1/s] | [m$^2$/s$^2$] | x10$^{-5}$[1/s] |
| Coast | 0.111 | 2.3 | 0.06 | 1.4 | 0.051 | 0.9 |
| Coast extension | 0.255 | 2.9 | 0.104 | 1.4 | 0.152 | 1.5 |
| Retroflection | 0.256 | 2.2 | 0.171 | 1.3 | 0.085 | 0.9 |
| Cape Cauldrion | 0.072 | 1.3 | 0.047 | 0.7 | 0.025 | 0.6 |




Pulses propagate downstream from Box 1. Figure 4 showed that these small-scale, rapid dynamics have speeds of 20 km/day on average, similar to the sub-mesoscale barotropic anomalies generated on the Agulhas plateau as described by Ruijter et al. (1999), Krug et al. (2017) and Tedesco et al. (2019). They are weakly present in the early period of the simulation, but their amplitude increases during the 3 months in autumn following the passage of the first Natal Pulse. During the following winter-spring months (July-October), the dynamics return to a less energetic state in this Box 2, as this barotropic instability occurs

when the Agulhas Current is not in a meandering state (Tedesco et al., 2019). This critical small-scale adjustment process is undetected in today's altimetric maps.

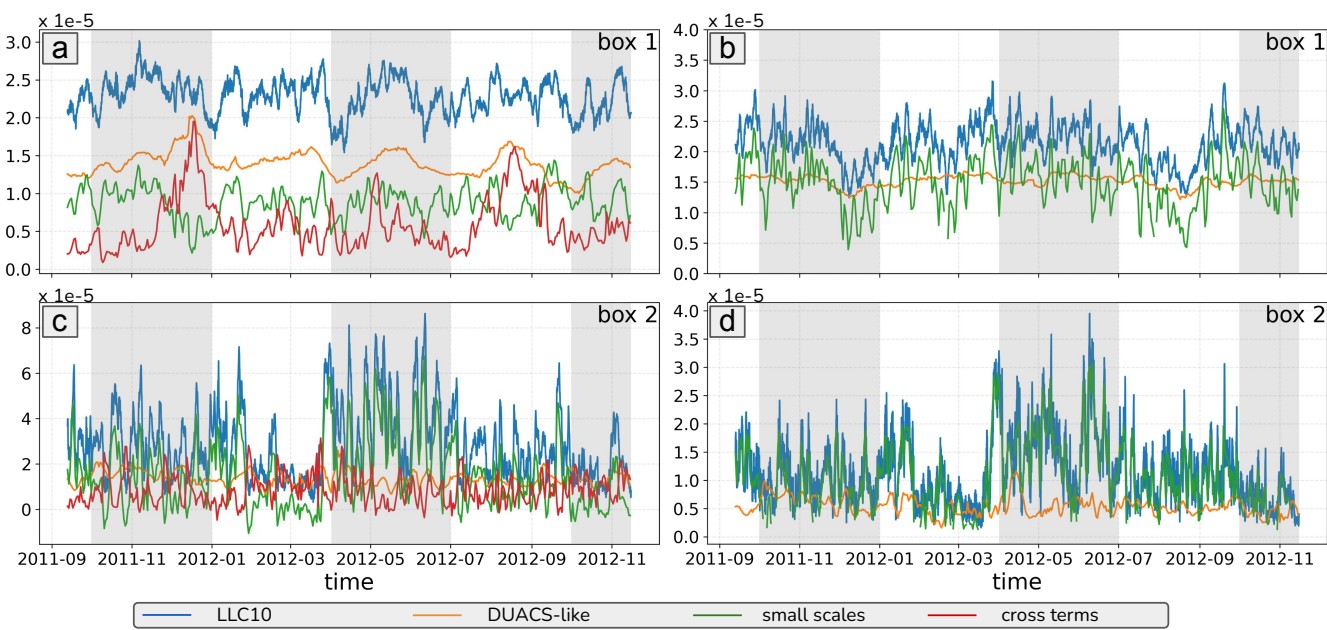

**Figure 6.** Time series of geostrophic strain rate [1/s] in the two coastal boxes for the full LLC10 simulation (blue), the pseudo-DUACS data (orange), the residual high-frequency smaller-scale (green), and the large/small scales cross terms (red). The left column represents the time series of the spatial average over box 1 (top) and box 2 (bottom), and the right column is the time series of the spatial std over the same boxes

To complete the analysis we focus on the flow's deformation with the potential of generating secondary vertical overturning circulation at different scales (Zhang et al., 2019). Figure 6 shows the geostrophic strain rate variability over the two of the boxes (box 1: a-b, box 2: c-d). The left column shows the spatial average of strain over the box, whereas the right column

shows the spatial std. This figure and Table 1 confirm that the larger-scale pseudo-DUACS geostrophic strain rate contributes around 60% of the total mean LLC10 strain rate in Box 1 (and in boxes 3 and 4, not shown). Whereas the residual small-scale rapidly evolving geostrophic strain contributes 40% of the mean strain rate, but most of the variability.

The Natal Pulses, the strongest events in terms of EKE, have only a small signature on the strain, especially on the pseudo-DUACS strain. They propagate through the region (a, b, c, d) with very high SSH and EKE values but with a less strong



impact on the deformation of the flow: the very large EKE peaks in December 2011 and August 2012 in box 1 (Figure 5a) have moderate DUACS strain peaks (Figure 6(a, b)), but other periods with weaker EKE also have moderate pseudo-DUACS strain peaks in box 1, of the same order of magnitude. Indeed, the passage of the Natal Pulses tends to damp out the smaller-scale strain (evident in the std, Figure 6(b)), but the total strain in Box 1 remains fairly constant throughout the year, with a mix of larger and smaller-scale contributions. In contrast, the effect of the Natal Pulse is evident on the cross terms (red line in Figure

6a ), whose effect increases during the pass of the Natal Pulses and causes the dampening of the smaller scales.

     An exception is Box 2 (Figure 6b,c)) where the small-scale strain is the most active all year, except when the Natal Pulse passes. Small-scale strain remains present even during the low EKE period (August to October 2012), with implications for vertical overturning and mixing at small scales in this region. In contrast with Sasaki et al. (2014), there is no evidence during this one-year simulation of small mesoscale EKE in winter/spring feeding energy to larger mesoscale EKE in late

spring/summer. Rather, the simulation appears dominated by more individual eddy events.

### 3.3    Geographical distribution of the energy cascade

One of the main aims of SWOT will be to understand and monitor a wider range of mesoscale structures, whether they act to reinforce or compensate for the larger scale variability, and to observe the interaction and transfers of energy between the different spatial scales. A direct (or downward) cascade is defined as the transfer of kinetic energy towards smaller scales. An

inverse (or upward) cascade is the transport of kinetic energy to progressively larger scales. We analyze the geostrophic energy fluxes in our region using the coarse-graining method, as displayed in Figure 7. Note that though the SSH observed by SWOT includes both geostrophic and ageostrophic components that dominate submesoscale features, only the geostrophic ones are accessible and can be used for computing energy transfers.

     In this analysis, our subsampled model at 1/10° around 45° in latitude has a Nyquist sampling of around 15 km, in line

with the SWOT observability (not far off 15 km) from Section 4. Since the objective is to perform a comparison with the pseudo-DUACS products, Figure 7 shows the geostrophic energy flux at 60 km (top) and 150 km (bottom) for the LLC10 data (a, c), and for the pseudo-DUACS product (b, d). Shorter wavelengths would not be representative of the pseudo-DUACS data. In line with Schubert et al. (2020), the inverse cascade's (blue) strength and spatial distribution increases at the larger 150 km scale. The full "SWOT-like" LLC10 presents a stronger inverse cascade, and the change of sign between upscale and

downscale fluxes occurs at shorter wavelengths. This has already been observed by Renault et al. (2019) in the Agulhas current and the Gulf Stream and indicates that current altimetric maps are unable to accurately estimate the spatial scales at which the energy cascade changes from an inverse to a direct cascade. The "patched" shape of the flux may be due to the limited duration of our model: only one year of data may not be long enough in this active region to provide stabilized mean fluxes, especially given the very energetic and diverse events shown in Figure 4. Both Schubert et al. (2020) and Contreras et al.

(2023) use a multi-year simulation for their flux computation which shows an inverse cascade dominated by the balanced flow, becoming stronger as larger wavelengths are analysed. At 150 km, the zone of the direct cascade is larger than at 60 km, and it is concentrated east of Port Elisabeth (around 26° E). It corresponds to the zone and wavelengths where Krug and Penven (2011) found the generation of smaller eddies after the passage of the Natal Pulses, in relation with the downstream decrease





in the number on Natal Pulses south of Port Edward (30°E, 31°S). In Figure 5a the Natal Pulse propagating downstream from
the Natal Bight breaks into many smaller and faster dynamics of Figure 5b creating a direct geostrophic energy cascade. As
seen in the time series, this very large and energetic event is the only one that is fairly well represented by the pseudo-DUACS
product, although the inverse cascade is largely underestimated.

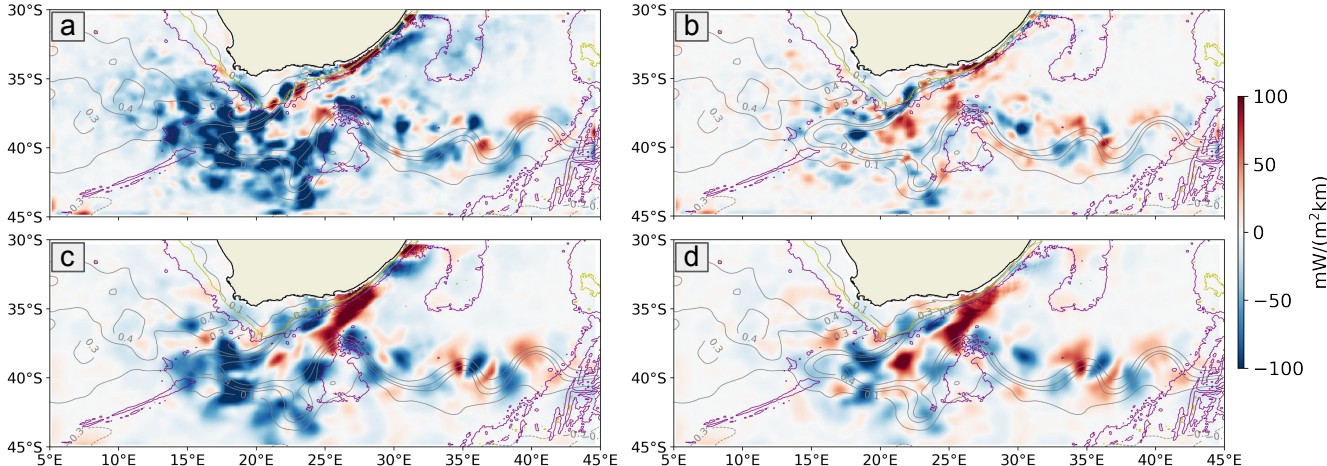

**Figure 7.** Geographical distribution of the yearly average (09-2011 to 11-2012) direct (red) and inverse (blue) energy cascade in the Agulhas
region computed with the coarse-graining method for 60km (top) and 150km (bottom) scales. (a, c) are computed with the LLC10 data, and
(b, d) with the pseudo-DUACS data. Grey lines are the mean SSH contour, and purple and yellow lines are the 3000m and 1000m bathymetry
respectively

This coarse-graining diagnostic will not be reproduced at the maximum resolution on the SWOT tracks, because there is not
enough data available across the two 50 km wide swaths during the 1-day orbit revisit.

## 4   Eddy diagnostics observed by SWOT


The EKE and strain rate diagnostics have been computed on the swaths generated with the SWOT simulator, by interpolating
LLC10 onto the 2 km regular SWOT grid. The aim is to understand how these diagnostics will be observed with SWOT data,
how much we will lose in terms of swath width, and how the residual instrumental error will impact the observability of the
SSH and the eddy diagnostics derived from it.

The strain rate and the EKE are both invariant with respect to the system coordinates (see Appendix B). So we can compare
these diagnostics from the 2D LLC10 maps computed with the geostrophic velocities in the zonal-meridional framework, with
similar diagnostics derived from SSH on the swaths computed in the along/across-track framework.



## 4.1 Impact of the SWOT instrumental noise

The objective of this section is to analyze the impact of KarIn noise on the SSH measurements and therefore on the derived
eddy diagnostics. We also infer the capability of the U-net noise mitigation technique, trained on the eNATL60, to treat a
different model (LLC10) in a different and very energetic zone. After denoising, we expect to be able to see the smallest
possible ocean structures. Gómez-Navarro et al. (2018); Chelton et al. (2022); Tréboutte et al. (2023) showed that KarIn's
random error adds noise to the signal features that could lead to a misinterpretation of the observed variable, or completely
mask the signal making it impossible to interpret the data. The impact of the small-scale random noise increases when making
first or second order spatial derivatives, such as the EKE or the strain. Figure 8 demonstrates this point, showing the SSH
(left), EKE (centre) and strain (right) in the ideal, LLC10 model scenario (top), when random noise is added (centre) and after
U-net is applied to remove the noise (bottom). All plots refer to pass number 5 (geographically shown in Figure 1). The noisy
SSH (b) presents the same large-scale features with respect to the simulated true (a), but smaller-scale features are introduced
or modified, leading to a misrepresentation of the SSH variability at small spatial scales. Whereas the noisy EKE (e) only
shows the strongest feature in the middle of the swath, and the strain (h) is completely covered by the noise, preventing any
interpretation of this variable. The U-net noise mitigation technique, with parameters derived from a different model in the
North Atlantic, suitably restores the input signal in the three cases. For the SSH, for wavelengths between 15 and 50 km, the
noise is reduced by one order of magnitude. However, U-net also removes part of the signal by a few percent points over
this interval. Thus, the ocean SSH dynamics observed after noise reduction will have the correct positioning and phase, but
their amplitude is slightly underestimated. The interval in which the signal is slightly underestimated is larger for higher-order
derivatives, being between 10 and 100 km for the strain.

The scores used to assess the impact of U-Net are defined in Section 2.3.2. The RMSE is shown in Figure 9. A synthetic
overview of the overall noise mitigation efficiency in the Agulhas current zone with the MITgcm model is given in Table 2,
comparing the scores and observability of this study to the original U-net simulation based on the eNAtl60 North Atlantic
study (Tréboutte et al., 2023), and the use of the filter implemented by Gómez-Navarro et al. (2020). The results are promising,
with an homogeneous RMSE for the four daily-sampled tracks in the Agulhas region, ranging between 0.20 cm and 0.45 cm,
which is in line with the findings from Tréboutte et al. (2023) for the North Atlantic. The RMSE across-track shape reflects the
simulated random error that increases towards the inner and outer swath borders (Esteban-Fernandez, 2013).

## 4.2 SWOT observability

Ocean observability is defined as the wavelength for which the NSR equals one, meaning that the noise and the signal have
equivalent energy. Following Tréboutte et al. (2023), for noisy fields where the error is always more energetic than the signal
(NSR always higher than one) we consider the observability to be larger than 1000 km. Previous findings by Wang et al. (2019)
show that SWOT observability is expected to be degraded and resolve wavelengths up to 35-45 km at high latitudes due to
SWH-induced instrumental noise. This is especially true in the Antarctic Circumpolar Current (ACC) where the wave-induced
random error has a large seasonality. In their simulation, Wang et al. (2019) used the realistic WAVEWATCH III (WW3) model



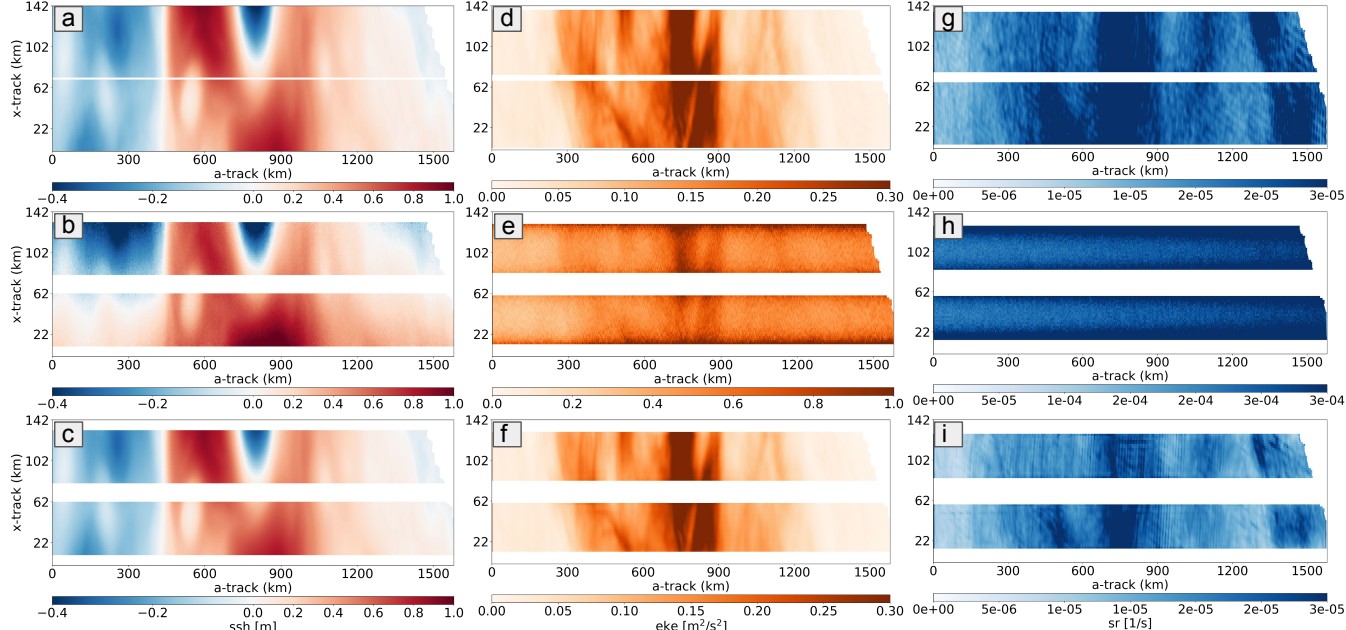

**Figure 8.** SSH (left), EKE (centre) and strain (right) in the ideal case of the non-noisy field (top), noisy (centre), and after noise mitigation with the U-net method (bottom). All plots refer to pass number 5 in Figure 1. The swath is represented horizontally with its distances along-track (a-track) and across-track (x-track) in kilometres where. The SSH is a snapshot of cycle 112, on January 1st 2012. EKE and strain refer are averaged over three months, simulating the CalVal scenario (January - March 2012). The left side of the swath corresponds to the southern section of the track, and the right side reaches the South-African coast

as input data for waves, and show that spatial filtering of the data to smooth the noisy smaller wavelengths destroys much of the information contained in the small-scale SSH field. Figure 10 shows the LLC10 SSH noise-to-signal ratio, before and after applying the U-net method over our four SWOT tracks. Even though we are using a noise simulated for a 2m wave field, as in Wang et al. (2019), the wavelength SWOT should resolve in this region before denoising is 40 km, whereas when U-Net is applied to the noisy data, an observable wavelength of 17.5 km can be restored. The resolved observable scale is also calculated on the EKE and on the strain rate. For noisy data, the observability decreases by a factor of three for first-order derivatives such as EKE, resolving scales of more than 120 km with noisy data. For second-order derivatives such as geostrophic strain, the noise dominates the signal for all wavelengths, since the NSR spectrum is always > 1. After U-Net application, the observability is restored to wavelengths of the order of 15 km in both cases.

Considering the results for these de-noising scores, and the comparison with other noise reduction techniques used in the literature, the U-Net method is very promising for the Agulhas current when analysing a realistic simulation. In terms of SSH mean RMSE, without any noise treatment, the values in the Agulhas and in the North Atlantic are comparable, 1.17 cm and 1.27 cm respectively. Using the U-net method in the Agulhas, the final RMSE is 0.24 cm, which is the same found by Gómez-Navarro et al. (2020) with specific tuning of their algorithms for the North Atlantic. The U-net performance in the North Atlantic has

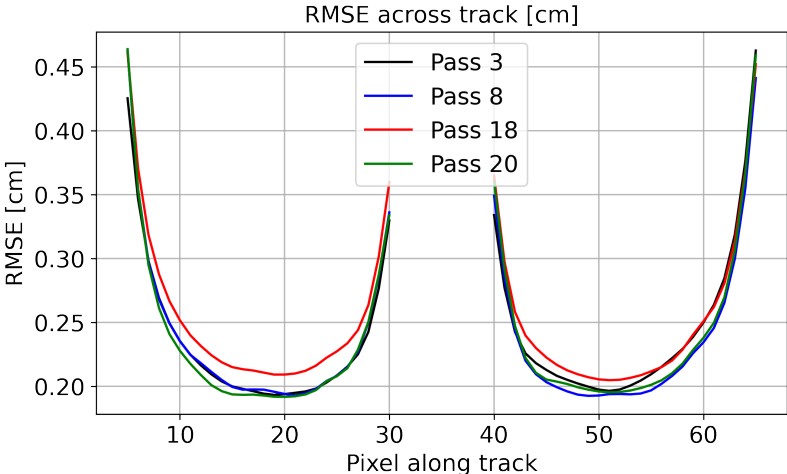

**Figure 9.** SSH root mean square error for each of the SWOT one-day CalVal tracks in our zone, averaged over three months (January-March 2012)

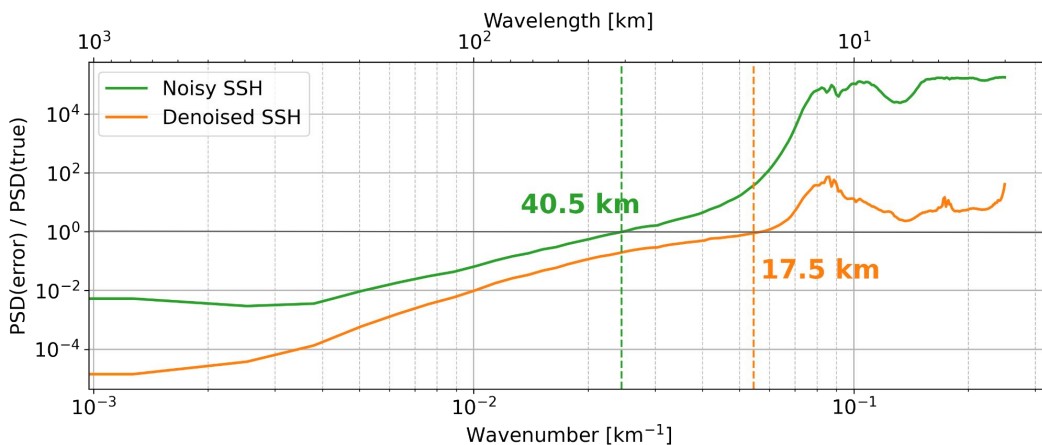

**Figure 10.** Estimating the NSR as the ratio of the power spectral density of the SSH error compared to the full LLC10 SSH fields for two cases: in green, is the spectrum of the noisy SSH, and in orange is the spectrum of the U-Net treated SSH. The vertical dashed lines indicate the wavelength at which the signal reached the noise (NSR=1). The PSD of the error and the noise are calculated for along-track PSD over the 1600 km segment and averaged for each across-track position, and for each track and cycle

an even better error reduction of 0.19 cm. In terms of the variance of the SSH residuals, the results are similar, with the best result found with U-net in the North Atlantic, the U-net in the Agulhas and the Gomez filter in the North Atlantic having comparable values. This result was expected because the Gomez filter was parametrized specifically on their study zone, and the U-net training was performed in the North Atlantic with a different model. What is remarkable about U-Net performance is that even with a different model and in a different zone, the mean RMSE and variance of SSH residuals are comparable





**Table 2.** Summary of the main noise reduction and observability scores for the two cases of the North Atlantic and the Agulhas regions. North Atlantic de-noising is computed in the two cases of the U-net method, and the filter from Gómez-Navarro et al. (2020). The Agulhas current has been treated only with the U-net method

|  | Agulhas region | | North Atlantic | | |
|---|---|---|---|---|---|
|  | No filter | U-net | No filter | U-net | Gomez |
| **SSH RMSE [cm]** | 1.17 | 0.24 | 1.27 | 0.19 | 0.24 |
| **Variance of SSH res [cm$^2$]** | 1.5 | 0.06 | 1.63 | 0.04 | 0.07 |
| **SSH wavelength [km]** | 40.5 | 17.5 | 42 | 10 | 27 |

with the original parametrization of the other methods. The SSH observable wavelengths give the most surprising result in this sense. The two zones have comparable observability before noise reduction (40.5 km in the Agulhas and 42 km in the North Atlantic). The U-net in the Agulhas manages to retrieve wavelengths down to 17 km. This result in the Agulhas could be improved in the future by performing new training in this specific zone. This is encouraging for the early SWOT processing, since the U-Net parameters, trained in the North Atlantic, could be applied directly to the early SWOT data, and then improved

by retraining U-Net once enough SWOT data has been retrieved. Finally, this study was conducted in preparation for the real SWOT data that will represent a new challenge, since in the early months of the mission's data analysis, no training dataset will be available for the U-Net algorithm. Note that in all the noise mitigation techniques presented in Table 2, the simulated random noise is estimated in the same way, based on the SWOT Project's spectral best estimates of the random noise for 2 m SWH (Gaultier et al., 2016). In reality, the SWOT random noise will vary in amplitude and could be impacted by small-scale

anomalies that will need extra editing (eg presence of isolated ships, icebergs, platforms, extreme waves, . . . ). However, the U-Net performance on a different region and model from the original training is very encouraging, not just for the restored SSH, but also for the key high-order eddy diagnostics such as EKE and strain.

## 5 Conclusions

Our study addresses the Agulhas Current mesoscale dynamics and its observation with current altimetry maps and with the

new 2D SAR-interferometric swath sampling brought by SWOT. We investigated how eddy diagnostics can be observed with a synthetic pseudo-DUACS dataset reconstructed from the LLC10 simulation data, and how SWOT will change the current paradigm by observing new space and time frequencies. Finally, we studied the effect of instrumental random error on SWOT observations and quantified the potential of a new de-noising algorithm based on a neural network approach.

To reproduce altimetry-like data from our model, we corrected the LLC10 SSH for the barotropic tide and the DAC as a

standard altimetric data processing step, and then calculated the geostrophic currents from the corrected SSH. Some high-frequency signals remain in the corrected SSH fields, due to residual barotropic tides, internal tides, and internal gravity waves (see Figure 2). These are minimised in our analysis since we are dealing with daily averaged model data for the pseudo-DUACS



and therefore with the daily residuals of small-scale fields. In reality, SWOT will fly across this region at 7 km/s (passing over the 15° in latitude in 4 minutes) and so during the CalVal phase, SWOT will measure daily 2D snapshots including these

residual high-frequency signals. To verify the impact of this on our eddy diagnostics, we compared the daily averaged EKE derived from the full LLC10 fields versus a daily snapshot from the hourly LLC10 model data. In the high-energy Agulhas region, the differences were minimal. Even in the area of higher internal tides radiating westward from the Benguela Current, the impact on the spatial derivatives of SSH such as EKE was small. The Agulhas Current is one of the most energetic regions of the global oceans, and other less energetic regions may need to use specific internal tide corrections (Zaron and Ray, 2017)

or more sophisticated techniques to separate the high-frequency internal gravity wave field from the rotation-dominated eddy fields (Le Guillou et al., 2021).

The separation between large scales (>150 km) from the DUACS-like reconstruction and the residual small scales (<150 km) with the total LLC10 simulation data has allowed us to quantify the EKE that is missing in today's observations during strong events like the Natal Pulses. In the 2011/2012 period, LLC10 shows two Natal Pulses flowing southwards from the

Natal Bight to Port Elizabeth. These large structures are well known and have been previously documented with altimetry and SST data (Krug and Penven, 2011; Krug et al., 2017). Our analysis shows that their location and phase are correctly estimated by the DUACS-like reconstructed observations, but we are missing 30-40% of the magnitude, due to the interpolation used in the mapping process. SWOT should also observe snapshots of the smaller and faster dynamics southwards of Port Elizabeth, which are mainly created by barotropic instabilities (Krug et al., 2017; Tedesco et al., 2019), and are totally missing in current

altimetry products due to their small diameters. Our EKE analysis also highlighted that the cross terms between the small and larger-scale dynamics are also important contributors to the total EKE field that are not observable today.

The Agulhas region is also characterized by strong mesoscale to submesoscale strain, estimated at 6 - 8 x $10^{-6}$ s$^{-1}$ by (Zhang et al., 2019) based on gridded altimetry estimates (their Figure may be saturated at its highest strain scales). Our modelled strain rate is a factor of 2-4 times larger, with mean values of 1.5 to 3 x $10^{-5}$ s$^{-1}$ for all scales. We showed that

although the modelled average strain rate magnitude is dominated by the large mesoscales in most regions, except in box 2 where the smaller-scale barotropic instabilities emerge, most of the strain variability occurs at smaller scales. Currently, available observations completely miss this information as the pseudo-DUACS data reveal a fairly constant and low strain variability in all regions and times of the year, even during the strongest changes in the EKE carried by the Natal Pulses.

Correctly assessing these dynamics is crucial because the Agulhas Current nearshore region has a rich upwelling system

(Blanke et al., 2009; Goschen et al., 2015; Jacobs et al., 2022). Small-scale and rapid changes in the SSH, EKE and strain as described here are capital for the stability and generation of local upwelling pulses and the associated rich biomass (Largier et al., 1992; Zhang et al., 2019).

The resolution of these smaller dynamics will be essential to correctly observe the ocean's energy cascade and in particular the inverse cascade which today is underestimated in magnitude and shifted to larger wavelengths (Renault et al., 2019). In

future SWOT measurements, the smaller-scale ageostrophic currents will not be accessible and even the geostrophic eddy cascade will have a spatial resolution of around 20km at best. Thus, the interaction between balanced and unbalanced motions, that at small scales Contreras et al. (2023) showed to be a key component for the knowledge of the forward cascade in the



Gulf Stream, will not be accessible. However, geostrophic balanced motions dominate for the SWOT-observable scales in both the Gulf Stream (Contreras et al., 2023) and in the Agulhas region (Schubert et al., 2020). Thus, within the scales of interest
for SWOT, we will not capture most of the inverse cascade, and the geostrophic contributions of the direct cascade ad scales larger than 15 km such as the barotropic instabilities observed near Port Elisabeth. The coarse-graining method will not be directly implemented on the SWOT swaths as the two 50 km wide swaths are not wide enough to correctly implement the 2D methodology used here. However, in 2024, a new gridded high-resolution 2D DUACS product will be implemented that will also include SWOT data. We expect this new product to be able to represent a wider spectrum of spatial scales, making it
possible to interpret the geostrophic energy fluxes between scales.

The results presented in this study are subject to how well the LLC10 simulation and the SWOT simulator represent reality. A few caveats are known for the model simulation, as discussed in section 2. The simulator also includes estimates of SWOT's systematic errors (satellite roll, phase errors, baseline dilation, timing errors, orbital errors), and the SWOT project is developing cross-calibration techniques to estimate and remove these additional errors (Dibarboure et al., 2022). Due to their nature, these
platform errors are expected to be significant only for wavelengths larger than 1000 km. However, high-frequency residuals could still be present at spatial scales shorter than 1000 km and therefore would translate into the SSH measurements and the higher-order derivatives. These residual errors, if present, would be subject to higher-level mission calibration against the available altimetry constellation (Dibarboure et al., 2022). Even after this cross-calibration step is applied, some residual errors may remain, and their impact on the derivation of eddy diagnostics needs to be addressed in future work. The main differences
with the real data are related to the generation and treatment of KarIn's random noise. Here a Gaussian distribution and a constant SWH of 2 m have been used to estimate the noise. The real SWH, however, can reach up to 7 m in the Southern Ocean, which would potentially produce a higher random noise and lower overall observability, and the noise may not have the same statistical distribution. However, early estimates by the SWOT Project suggest that the SWOT random noise may be smaller than in our simulations, unexpected and good news for future eddy observations. The U-Net methodology has benefits,
but it needs a training data set that will not be readily available for real data. However, our study shows that the parameters tuned for the eNAtl60 model (North Atlantic zone) are still providing good results in another region, period, and with data from another simulation that accounts for different ocean dynamics. Early SWOT data in 2023 will be in a different period, and we have to correct for residual errors without a training dataset. Based on the promising results of this work, the U-net noise mitigation technique can be applied to reduce small-scale noise, and it will be interesting to see the outcome and correct
potential differences.

**Appendix A:  Validation of the LLC4320 simulation**

The LLC4320 has been validated by numerous studies: Rocha et al. (2016, 2015) compared the model kinetic energy spectra in Drake Passage against long series in-situ ADCP data, finding good agreement both for the rotational and divergent components of the one-dimensional Helmholtz decomposition. Drushka et al. (2018) analysed the internal tide component of the 1/48°
MITgcm data and found good agreement with in-situ data from a glider. Wang et al. (2018) demonstrated that the simulation



reproduces well the location and amplitude of kinetic energy peaks, comparing the simulation to 25 repeats Acoustic Doppler Current Profiler (ADCP) surveys in the Northwestern Pacific Ocean. In the North Pacific Ocean, Savage et al. (2017) tested the performance of two global ocean simulations (the HYbrid Coordinate Ocean Model at 1/24° resolution and the LLC4320) to reproduce the diurnal, semidiurnal, and supertidal variance of the SSH against nine McLane profilers. Both models agree

on the diurnal and semidiurnal tidal ranges, but discrepancies were found at supertidal frequencies. The LLC4320 variance in SSH was found to be closer to the profilers variance because its higher spatial resolution allows it to better reproduce the energy transfer out of the inertial and semidiurnal bands.

  We estimate in Figure A1 the geostrophic EKE temporal mean for the LLC10, the observed DUACS all-sat product from AVISO, and the pseudo-DUACS product. In order to benchmark the realism of the model physics, we want to highlight the

large-scale circulation variability rather than the mean state by removing the annual cycle from the full data for the three cases (not shown). For the derivation of the EKE see Section 3. The time average over the full simulation period (09/2011 to 11/2012) shows that due to their smoothness, the real DUACS product and the pseudo-DUACS underestimate the EKE levels of energy with respect to the LLC10 simulation, while retaining the spatial features of the current system, in agreement with other studies (Chelton et al., 2019), in particular along the coastal Agulhas Current. The pseudo-DUACS underestimates the observed

DUACS/CMEMS reconstruction in the retroflection region (37-42°S; 15-20°E) (see Figure A1a and b), the mean EKE in the retroflection being 0.157 m$^2$/s$^2$ for the pseudo-DUACS and 0.178 m$^2$/s$^2$ for the real DUACS. Whereas the LLC10 has a mean EKE of 0.256 m$^2$/s$^2$. The mean axis of the coastal Agulhas Current, its retroflection around 20°E and the meandering Extension are more clearly delineated with the LLC10, and much more diffuse in the real DUACS maps. The LLC10 field has more energy at the retroflection south of 40°S and along the coastal Agulhas Current. The mean DUACS EKE in the Extension is slightly

shifted to the north with respect to the modelled LLC10 and pseudo-DUACS mean current. The coastal Agulhas current region has larger differences: at 33°S, the mean EKE in the DUACS maps and in the LLC10 are 0.039 m$^2$/s$^2$ and 0.111 m$^2$/s$^2$ respectively. Figure A2 represents a snapshot of the geostrophic surface relative vorticity, see Section 3, for the LLC10 and for the pseudo-DUACS product. This qualitative comparison demonstrates that the LLC10 model has energetic velocity gradients giving a good representation of small-scale vorticity structures. Whereas the reconstructed DUACS represents the amplitude

and the location of the main, large-scale vorticity structures but as expected, does not represent the small-scale vorticity in terms of rotational and strained flow. Since the LLC10 provides a good representation of the EKE and the small-scale vorticity, and since the objective of this study is to understand how SWOT will observe the variability compared to what is achieved today, the LLC10 is a good fit and has been used in our analysis. Furthermore, it has the advantage of consistent sampling with the additional fields used in this study: the pseudo-DUACS data (see section 2.2), and the SWOT cross-calibration data

mentioned in the discussion.

## Appendix B: Well-defined variables invariant to coordinate system rotation

In order to calculate the Eulerian diagnostics on a rotated along-track-cross-track field, we need to use *well-defined* variables. The full mathematical derivation can be found in Wirth (2015). In the following, we have a two-dimensional flow field com-



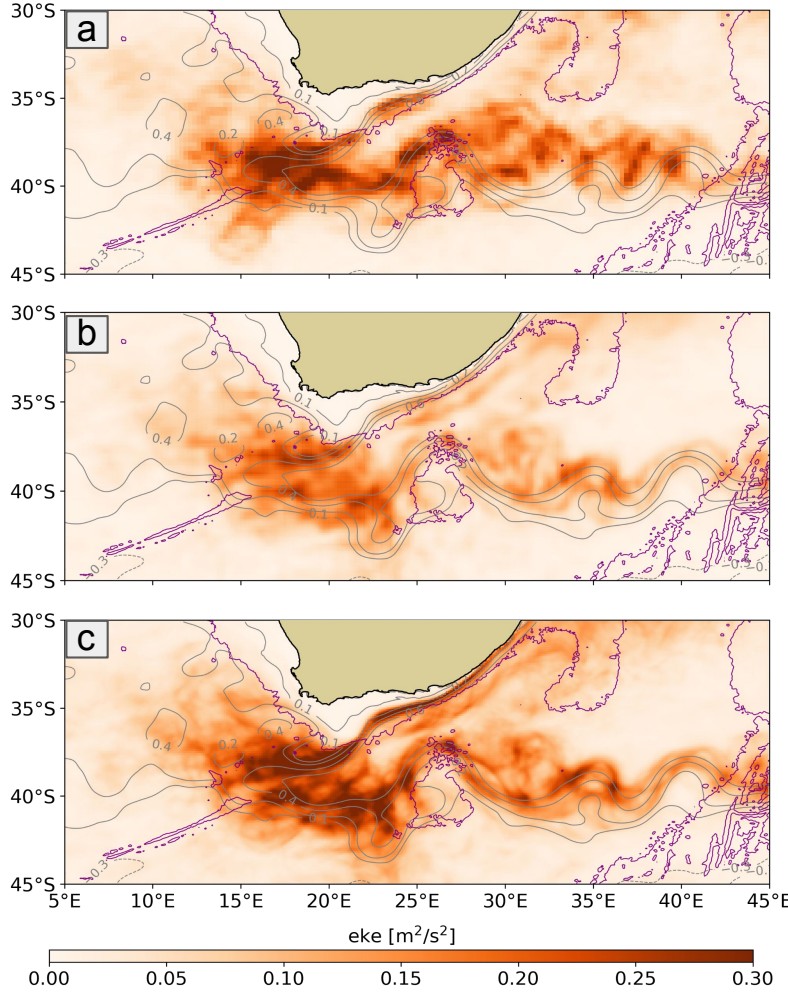

**Figure A1.** Geographical maps of EKE averaged over the period September 2011 to November 2012. Top: observed altimetry DUACS all-sat maps. Centre: pseudo-DUACS product. Bottom: LLC10. The grey contour lines represent the mean current. Purple lines represent the bathymetry at 3000m

posed by the velocity vector $(u(x, y, t), v(x, y, t))$. Some scalar quantities, referred to as *well-defined*, are invariant with respect to the coordinate system, which might be moved or rotated without any impact on the quantity's values. As an example, the speed is a well-defined variable, as opposed to the velocity vector, which changes when the coordinate system is rotated. The linear deformation of a fluid volume is characterised by the strain tensor. Let $u$ and $v$ be components of the velocity vector in a first reference frame in a two-dimensional flow, the strain tensor is defined as

$$(\nabla u^t)^t = \begin{pmatrix} \partial_x u & \partial_y u \\ \partial_x v & \partial_y v \end{pmatrix} \tag{B1}$$



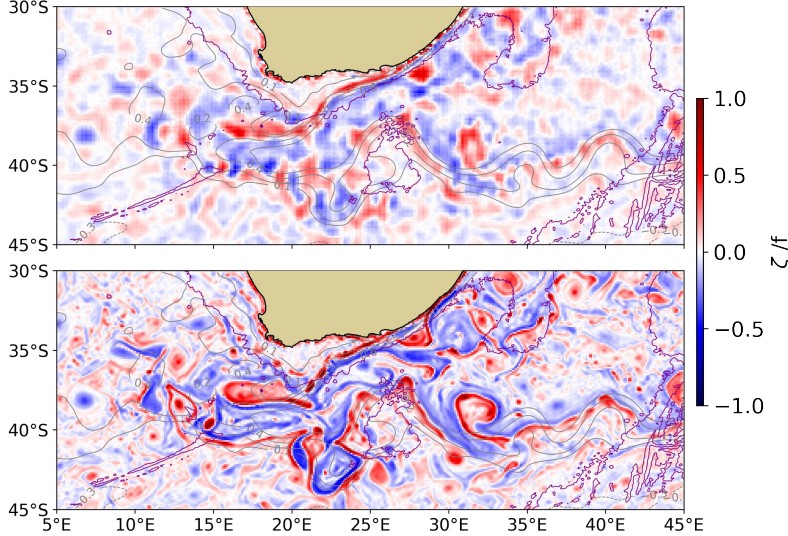

**Figure A2.** Snapshot of normalized surface relative vorticity, on January 1st 2012. Grey contour lines represent the mean current. Purple and yellow lines represent the bathymetry respectively at 3000m and 1500m. Top: pseudo-DUACS reconstructed data from LLC10. Bottom: LLC10

For a vector, the rotation to a second coordinate system is given by the rotation of the first system by an angle $\alpha$, by the left multiplication with the rotation matrix $A$

$$\begin{pmatrix} u' \\ v' \end{pmatrix} = \begin{pmatrix} \cos\alpha & \sin\alpha \\ -\sin\alpha & \cos\alpha \end{pmatrix} \begin{pmatrix} u \\ v \end{pmatrix} = \mathbf{Au} \tag{B2}$$

Its gradient transforms with the right multiplication with the transpose of the matrix A. Thus, the strain tensor transforms as follows:

$$\begin{pmatrix} \partial_{x'}u' & \partial_{y'}u' \\ \partial_{x'}v' & \partial_{y'}v' \end{pmatrix} = \mathbf{A} \begin{pmatrix} \partial_x u & \partial_y u \\ \partial_x v & \partial_y v \end{pmatrix} \mathbf{A^t} \tag{B3}$$


where $\mathbf{A^t} = \mathbf{A^{-1}}$. Note that several well-defined quantities can be derived from the strain tensor and depend linearly on it.

- – trace of tensor $d = \partial_x u + \partial_y v$

- – vorticity $\xi = \partial_x v - \partial_y u$

- – determinant $D = \partial_x u \partial_y v - \partial_y u \partial_x v$

– square of the components of the strain matrices $H = d^2 + \xi^2 - 2D$

- – strain rate $s^2 = d^2 + \xi^2 - 4D = H - 2D = (\partial_x u - \partial_y v)^2 + (\partial_x v + \partial_y u)^2$





- Okubo-Weiss parameter $OW = s^2 - \xi^2 = d^2 - 4D$

The strain rate is hence invariant with respect to the system coordinates. So is the EKE, derived from the speed. This justifies the comparison of the diagnostics on the 2D maps computed with the geostrophic velocities in the zonal-meridional frame, with the diagnostics on the swaths computed in the along/across-track frame.

*Author contributions.* Conceptualization and methodology, E.C, R.M, O.V.; software, E.C. and R.C.; writing, E.C., R.M., O.V; supervision R.M., O.V. and L.R. All authors have read and agreed to the published version of the manuscript.

*Competing interests.* The authors declare that they have no conflict of interest.

*Acknowledgements.* E.C was supported by the Centre National d'Études Spatiales (CNES) and Collecte Localisation Satellites (CLS) with a co-funded research grant to conduct her PhD. This work was financed through the CNES SWOT TOSCA Project.



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
