# Peer review of "Ocean 2D eddy energy fluxes from small mesoscale processes with SWOT"

_EGUsphere, 2023_

## Author Comment (AC1)

**Reply: 1st community comment**

**Major comments**

*1. Lines 260-290: The methods described here aim to show the potential of the SWOT mission to observe smaller mesoscale eddies (sometimes called larger submesoscale eddies) and the associated energy cascade. However, in addition to the instrumental noises, the SWOT-derived SSH at mesoscale also contain large signals of unbalanced internal tides and internal gravity waves. So, you can not directly calculate the geostrophic current and thus the EKE from the SWOT data based on geostrophic relation. Actually, how to remove the "noise" signals of unbalanced motions from the SWOT SSH data is still a large challenge for the community. So, I suggest you explain more clearly here to what degree can your data and methods represent the context and potential of SWOT data in reality.*

This is an important point for any analysis of alongtrack altimetry or SWOT 2D SSH data, but the relative importance of the internal tides and internal waves versus mesoscale eddy cascades varies from one geographical region to another. As mentioned in the text (lines 164-166), the Agulhas region has very strong mesoscale energy and cascades that largely dominate any internal tide signals at these relatively high latitudes. This is particularly the case for our statistical eddy diagnostics based on EKE or strain, which accentuate the shorter 2D space-scale dynamics. Indeed, since we have model fields, we could have separated the balanced and unbalanced motions analytically, e.g. with a Helmholtz decomposition or temporal filtering, as in Qui et al., 2018. However, our objective was to use the model's surface fields as Pseudo SWOT SSH data, and thus to not have the full temporal evolution of the high-frequency flow components. With this in mind, we made a careful analysis of our methodology and the impact of the high-frequency residuals in the zone we are studying. The second paragraph of the discussion treats this point (lines 504 - 516), and we have added a short point on these high-frequency residuals in Appendix C in the revised manuscript. A detailed description of our approach is given in the following.

a. We compared modelled geostrophic EKE and strain rate statistics computed from a) the full hourly time series, b) one hourly snapshot taken every day (eg. as in the SWOT 1-day orbit sampling), and c) the daily average. Each of these datasets represents slightly different high-frequency dynamics. The full hourly dataset represents the full dynamics, including the hourly evolution of the mesoscale/submesoscale structures, internal gravity waves and internal tides (barotropic tides and the Dynamical Atmospheric Correction have been removed). The daily snapshot represents only a fraction of the high-frequency dynamics <24h, and as such does not represent the full amplitude of the evolving coherent and incoherent internal gravity waves. The daily averaged data minimises the variations < 24h, and we use it as a proxy for the geostrophic motion. The top image shows the yearly std of the difference between the daily sampled and daily averaged data (std(x')). We see that the dominant differences are centred around the mean current, these are fast dynamics with a period <24h that we filter out when averaging the data over one day. These dynamics do not have the typical structure of internal gravity waves or internal tides, they could be due to the rapid movements of the energetic currents over time. Here we have imposed geostrophic balance over both slower and faster dynamics even though the latter may be due to rapid ageostrophic adjustments.

[Figure]

The central image shows the std of the EKE from the full hourly dataset over one year (std(x)). We can see here how well correlated the high-frequency differences are with the full std (EKE): it is dominated by high-frequency modulations of the most energetic currents (note the difference in energy level: the high-frequency variations < 24h are an order of magnitude smaller than the std (EKE).

[Figure]

The bottom image quantifies the variance explained between the low-frequency (LF) dynamics EKE (averaged over one day, dynamics > 24h) and the full hourly EKE:

$$\left(1 \; - \; \frac{var(LLC10_{EKE} - LF_{EKE})}{var(LLC10_{EKE})}\right) \; * \; 100$$

This tells us how much the low and high frequencies dynamics contribute to the total std. We see that the % variance is large everywhere, meaning that the full dynamics is mainly represented by the lower frequency dynamics, except on the shallow plateau and off the Benguela coast where higher frequency dynamics dominate. The white contours represent a mean EKE of 0.04 m$^2$/s$^2$ and 0.16 m$^2$/s$^2$ (ref Figure 3a). In the region of stronger EKE (>0.04 m$^2$/s$^2$), 7% of the total std is due to some HF dynamics that are difficult to separate from the mean dynamics clearly, but that do not have the typical structure of internal gravity waves or internal tides.

[Figure]

Variance explained - LLC10 and HF dynamics

b. In most of the regions analysed in this paper (Agulhas Current, Retroflection, Extension, Cape Cauldron), there is very little difference in the EKE or strain, whether we use hourly or daily snapshots or daily-averaged data. The exception is around 30-32°S in the Atlantic where a larger component of internal gravity waves and internal tides are present in the zone. When analysing geostrophic currents from SSH gradients, there would be a small contribution in the direction perpendicular to the wave propagation and very little parallel. So, even near 30°S, the signature of the internal gravity waves would be small in the cross-propagation velocities in EKE, and negligible in the velocity gradients which are then used to compute the strain rate. Snapshots of the LLC10 strain rate do not present any strong gradient or shape linked to internal gravity waves and internal tides. So, we don't expect this to be a dominant feature in our zone.

c. Of course, this might not be the case in the tropics or subtropics where the signature of internal gravity waves and internal tides is much higher, nor in regions of strongly incoherent tides. In this case, special treatments would be necessary to correct these contributions, as in Zaron and Ray, 2017 or in Le Guillou et al., 2021. In our region, Qiu et al., 2018 show that balanced motions are the dominant dynamic.

Old text
Line 511-512 : "To verify the impact of this on our eddy diagnostics, we compared the daily averaged EKE derived from the full LLC10 fields versus a daily snapshot from the hourly LLC10 model data. In the high-energy Agulhas region, the differences were minimal".

New text
"To verify the impact of this on our eddy diagnostics, we compared the daily averaged EKE derived from the full LLC10 fields versus a daily snapshot from the hourly LLC10 model data. In the high-energy Agulhas region, the differences were minimal (see appendix C)".

*2. Lines 307-315: The residual and smaller scales of EKE and strain rate you defined here (i.e. difference between the results of LLC10 and the DUACS) also contain the contributions of the cross terms. You should clarify this point more explicitly.*

Indeed, the cross terms are contained in the residual EKE and strain. Thank you for pointing out that this requires a more thorough explanation, we have updated the manuscript accordingly. The contribution of the cross terms is quantified in Figure 5 and Figure 6 which present the temporal series of EKE and strain rate over four different regions of the Agulhas system. The cross terms are present in both cases, but they never dominate the EKE or strain rate. You will find an in-depth answer to this point in comment 2 of the first anonymous review.

*3. Lines 393-395: Could you explain the possible reasons accounting for the difference between your results and those in Sasaki et al. (2014)? By the way, I also suggest the authors briefly discuss the generation mechanisms of the smaller mesoscale eddies (sometime called larger submesoscale eddies) here. Are they generated by mixed-layer baroclinic instability?*

The explanation for this difference may be due to the specificity of this zone, and that our simulation is only 1-year long, so the "seasonal" calculations are dominated by large eddy events. In fact, we are very close to the Agulhas retroflection and separation where, during the year, each season's dynamics can be dominated by the propagation of large-scale, transient events (Lines 328 - 346) as visible in Figure 4, Figure 5 and Figure 6. In periods where no Natal pulses occur, faster and smaller-scale dynamics are generated by barotropic instabilities (Lines 351 - 354) and can become very energetic (Figure 5b). In both cases, the lateral stirring is strong, and the seasonal mixed-layer depth remains shallow all year round, in contrast to the strong seasonal mixed-layer depth changes in the North Pacific (see climatological Figure below). So the seasonal dynamics in the Agulhas Retroflection are weak compared to mesoscale events and do not strictly follow the paradigm of mixed-layer instabilities, contrary to Sasaki et al., 2014. This point is clarified in the revised version of the manuscript.

[Figure]

Figure: Global MLD in summer and winter from a model (left) compared to observations based on Argo analysis. The model results are computed over the 1991–2000 period from the historical simulation, whereas the observations are adopted from de Boyer Montégut et al. (2004), after Tjiputra, et al (2012). (Geoscientific Model Development Discussions. 5. 3035-3087. 10.5194/gmdd-5-3035-2012).

Old text

Lines 393-395: In contrast with Sasaki et al. (2014), there is no evidence during this one-year simulation of small mesoscale EKE in winter/spring feeding energy to larger mesoscale EKE in late spring/summer. Rather, the simulation appears dominated by more individual eddy events.

New text

In contrast with Sasaki et al. (2014), there is no evidence during this one-year simulation of small mesoscale EKE in winter/spring feeding energy to larger mesoscale EKE in late spring/summer. Indeed, the seasonal averages in this one-year simulation are dominated by more individual eddy events. The seasonal mixed-layer depth remains shallow all year round in contrast to the strong seasonal mixed-layer depth changes in the North Pacific (see climatology in Figure 5 of Tjiputra et al., 2012). So seasonal dynamics in the Agulhas Retroflection are weak compared to mesoscale events and do not strictly follow the paradigm of mixed-layer instabilities, contrary to Sasaki et al., 2014.

**Minor comments**

*5. Section 3.3 Geographical distribution of the energy cascade: In addition to geographical distribution, it is also meaningful to discuss the seasonal variations. Does the energy cascade has a clear seasonal cycle due to the enhanced mixed-layer eddies in winter?*

We do not have a quantitative response to the question on enhanced mixed-layer eddies in winter since we have only analysed the surface fields. This would indeed be an extremely interesting point to study in the future. However, we do not find a strong seasonal cycle by averaging the surface energy cascade geographical field over two different seasons (Figure below: end of Summer - top - and end of winter - bottom). As explained in comment 3, the seasonal cycle is weak in this region in contrast to the strong large mesoscale eddy events such as the Natal Pulses that tend to dominate any seasonal average, particularly since our model simulation is only 1 year long.

---

## Author Comment (AC2)

**Reply: comments 1st Reviewer**

*1. Line 301: "mean surface geostrophic EKE over this 1-year period," One year is a bit short for a mean surface geostrophic EKE here. This is why EKE pattern in Figure 3 is quite patchy. ... but I think this is the limitation of using LLC10 ?*

Exactly, this is the main limitation of our 1-year simulation, whether at 1/48° or 1/10° since the region is dominated by large mesoscale events with time-scales of 1-3 months that dominate any seasonal average and can impact on the annual average. However, SWOT's fast sampling phase will only be a three-month-long temporal series. And although it occurs in Apr-June, it will not be the seasonal aspect of this time period of interest, but the large mesoscale structures to be observed during this period. Understanding SWOT capabilities is really our main focus in the paper, to be aware of the potential and the limitations of the data we will analyse.

*2. Figure 5: What is the definition of the cross-terms?*

The total LLC10 EKE is not simply given by the addition of the individual contributions from the residual small scales EKE and from the EKE estimated with the pseudo-DUACS data. An additional cross-term arises representing the interaction between the large scales and the smaller-scale motions. The same is true for the strain rate. The analysis shows (Figure 5 and Figure 6) that in any of the boxes, the cross terms represent a minor component of the dynamics. We have added an expanded EKE equation and the following short text after the standard EKE equation (4), inserted before line 275 (old text):

"We note that when separating our EKE analysis into larger-scales represented by CMEMS mapping, and residual smaller scales, this EKE equation needs to be expanded and important cross-terms result, see eq.4a, that mix the large and small scales. These will be analysed in our results section."

We have also added an explanation in the strain rate calculation:

Old text: lines 277-278 "Its formulation is defined from surface geostrophic velocities. "

New text: "Its formulation is defined from surface geostrophic velocities. As with the EKE analysis, an expanded version of the strain rate is needed when calculating the difference between large CMEMS scales and residual small scales, which also include cross-terms. This component is analysed in the results section."

*3. Table 1: Values of strain for PseudoDUACS are quite large compared to Figure 3. Is this due to the specific locations of the boxes?*

The large values for the strain rate in Table 1 in comparison with Figure 3 are due to what these two represent. Figure 3 shows the std of strain over time in a 2D geographical map. It shows that most of the strain rate variability in the Agulhas region is due to the residual small scales and that current altimetry is not able to fully represent this variability. Table 1 shows the average strain rate value over the specific box area over time. This is highlighting the fact that even though most of the strain variability is carried by the small scales, around 60% of the strain amplitude is still carried by the larger scales, which are quite constant over the year. This is also visible in Figure 6a where the orange line representing pseudo-DUACS carries most of the strain rate's strength, but the green smaller scales in Figure 6b carry

most of the variability over the year. Thank you for pointing out that this is not clear, I have changed the manuscript accordingly:

Old text:
Lines 380-382: This figure and Table 1 confirm that the larger-scale pseudo-DUACS geostrophic strain rate contributes around 60% of the total mean LLC10 strain rate in Box 1 (and in boxes 3 and 4, not shown). Whereas the residual small-scale rapidly evolving geostrophic strain contributes 40% of the mean strain rate, but most of the variability.

New text:
This figure, together with Table 1 which quantifies the average strain rate values over the boxes and over one year, confirms that the larger-scale pseudo-DUACS geostrophic strain rate contributes around 60% of the total average LLC10 strain rate amplitude in Box 1 (and in boxes 3 and 4, not shown). Whereas the residual small-scale rapidly evolving geostrophic strain contributes 40% of the mean strain rate, but most of its variability as shown in (Figures 3d, 3f, 5b, 5d).

***4. Figure 7: if the figure is dominated by the propagation of Natal Pulses, one possibility would be to do the statistics in absence of Natal Pulses (as in Tedesco et al., 2019).***

Indeed, this would be an interesting comparison if we had a longer model simulation to derive more robust energy cascade statistics of the period between Natal Pulses. Tedesco et al (2019) used a 1-month and a 5-months high-resolution simulation with no Natal Pulses to perform a robust study of submesoscale frontal eddy generation. As shown in Figure 4, we lose around 7 months of data if we remove the Natal Pulses from Figure 4 in between box 1 and 2. The remaining data would not be enough for our monitoring of the effect of mesoscale structures over the larger scale variability, for which Schubert et al. 2020 and Contreras et al. 2023 use muli-year simulations.

It's still interesting how the direct cascade dominates from the mix of Natal Pulses and smaller-scale instabilities off Port Elisabeth, and the LLC10 shows the clear geostrophic energy cascade from the Retroflection and further west. Frankly, we were pretty happy!

---

## Author Comment (AC3)

**Reply: comments 2nd reviewer**

*1. L210: Systematic errors such as satellite roll, and phase errors, could be much bigger sources of error on SSH than Karin random error. Can the authors elaborate on the reason why they choose to focus on Karin random error on SSH data and observable wavelengths in SSH, eddy kinetic energy, and strain rate?*

You are correct, systematic errors can add >20 cm to the SWOT SSH signal. However, the SWOT Project's simulations demonstrate that 1) we are efficient in the removal of these contributions (Dibarboure et al., 2022), and 2) These simulated systematic errors affect wavelengths that are larger than the mesoscales observed in this manuscript, so the impact on our statistics is minimal. Karin random error is the key component for the small-scale gradients observed here. We have added a sentence on this in our SWOT simulator section.

Old text:
Line 209: "Dibarboure et al. (2022) estimate that the systematic errors alone contribute tens of centimetres in SSH. "

New text:
Dibarboure et al. (2022) estimate that systematic errors alone contribute tens of centimetres in SSH. However the SWOT Project's cross-calibration techniques are efficient in reducing these simulated systematic errors, and since their wavelengths are larger than the mesoscales observed in this manuscript, the impact on our eddy statistics is minimal.

*2. L391-L395 The paper's findings don't seem to align with Sasaki et al. (2014)'s research regarding the energy feeding dynamics between different mesoscale EKE during various seasons. If the paper doesn't adequately address this discrepancy, it could be a point of critique.*

The explanation for this difference may be due to the specificity of this zone, and that our simulation is only 1-year long, so the "seasonal" calculations are dominated by large eddy events. Our analysis shows that seasonal dynamics in the Agulhas Retroflection are weak compared to mesoscale events and do not strictly follow the paradigm of mixed-layer instabilities, contrary to Sasaki et al., 2014. For further detail please refer to our response to the Community comment 3. This point is also clarified in the revised version of the manuscript.

*3. L233-244 U-Net's General Application: The authors show that U-net performs well even with different models and in different zones, leading to similar Root Mean Square Error (RMSE) and variance of SSH residuals. This indicates the potential universal applicability of the U-net method across various regions and circumstances. While the authors show U-Net's promise, the discussion lacks a thorough analysis of why U-Net performs similarly across various regions and models. It's not clear how U-Net achieves this consistency, which might lead to skepticism about the broad application of U-Net in other environments. Moreover, the potential improvements that could be achieved by training U-Net specifically for different regions could be discussed in more detail.*

The U-net works on data statistics and aims to minimise the gap between the noisy ssh and the simulated noise. Its performance is best in its training domain, which is the case for the noise added to the NATL60 with the SWOT simulator which is well-characterised.  One key point is that the SWOT simulated noise that is added to the MITgcm is based on the same spectral estimates as those used for the U-net training: it is not regional, it is a global spectral estimate. The noise mitigation from another training region is thus quite efficient. On the other hand, if the input SSH is too different from the one used for the training, the

performance will decrease. For this, the NATL60 SSH used for training and the MITGCM SSH before noise mitigation are normalised with respect to their variance and mean. Then once the noisy SSH is mitigated with the U-net methods, the normalisation is reapplied to find a SSH similar to the input one.

*Now, the noise on the real data may be very different from the simulated one, so we need to recharacterize the noise on the real data. Then, we will need to simulate new SSH with the actual noise and retrain the U-net with this new dataset for efficient random noise mitigation.*

We have added a more detailed discussion on this in the final paragraph of the discussion: see response to comment 5 below.

**4. L383 The study found that Natal Pulses, despite being the strongest events in terms of Eddy Kinetic Energy (EKE), had only a small impact on the strain, particularly on the pseudo-DUACS strain. This could be criticized if it contradicts the existing understanding of the effects of Natal Pulses on ocean dynamics or if the paper does not provide a sufficient explanation for this finding.**

This is an interesting point. To our knowledge, the effect of Natal Pulses on an ocean strain field has not previously been investigated. However, Natal Pulses are large, slow-moving structures compared to the small scales. So although they have a strong SSH signature in their core, and induce gradients and currents only at their edges, they evolve more slowly in time. Figures 5 and 6 highlight that in Box 1, the large-scale CMEMS strain rate and the cross-term strain both increase slightly as the Natal Pulse passes, but this is compensated by the smothering of the small-scale strain. This probably warrants a more detailed study in some future work, maybe with real SWOT data.

**5. SWOT Observability: The authors employ the U-net noise mitigation technique and assess its impact on different models. Despite the efficacy of the U-net method, it does slightly underestimate the signal amplitude. Also, there is no clear discussion on the limitations of using the U-net technique and the reasons behind its efficacy.**

Thank you. We have added some of these points to the final discussion:

Old text
Lines 565-570: The U-Net methodology has benefits, but it needs a training data set that will not be readily available for real data. However, our study shows that the parameters tuned for the eNAtl60 model (North Atlantic zone) are still providing good results in another region, period, and with data from another simulation that accounts for different ocean dynamics. Early SWOT data in 2023 will be in a different period, and we have to correct for residual errors without a training dataset. Based on the promising results of this work, the U-net noise mitigation technique can be applied to reduce small-scale noise, and it will be interesting to see the outcome and correct potential differences.

New text:
Finally, the U-Net noise mitigation is very promising. The technique slightly reduces the signal (by a few percent) but does not over-smooth the gradients, and retains the main structures and anomalies up to the coast. This is a strong benefit for eddy diagnostics and provides better noise mitigation with respect to the other filters analysed by Treboutte et al. (2022). Our study shows that the U-net trained for the eNAtl60 model (North Atlantic zone) is still providing good results in the Agulhas region in a different period, and with data from

another simulation that accounts for different ocean SSH dynamics. One of the reasons for the U-Net efficiency is that the U-net was trained in the North Atlantic for different wave heights and seasons trying to be as representative as possible for dynamics in other regions. Besides, the simulated SWOT random errors are based on global spectral estimates and are not regionally varying. This may not be the case for the early SWOT data in 2023, with potentially a geographically-varying random error. In applying the U-net technique, we also take care that the input SSH is not too different from the one used for the training, to maintain a good performance. For this, the training SSH and the SSH are both normalised with respect to their variance and mean before the U-net technique is applied, and then the inverse normalisation is applied to recover the correct input SSH. One of the disadvantages of the U-Net method is that it needs a solid training data set that will not be readily available for the early SWOT data. However, the promising results of this Agulhas study, based on a different model and region, are very encouraging. The parameters derived from the U-net technique trained on simulated data and noise may be used for the first random noise correction for SWOT, and it will be interesting to see the outcome and correct for potential differences when applying it to the real data.